



**Ice multiplication from ice-ice collisions in the high Arctic: sensitivity to ice habit, rimed fraction and the spectral representation of the colliding particles**

Georgia Sotiropoulou[1,2], Luisa Ickes[3], Athanasios Nenes[2,4] and Annica M. L. Ekman[1]

[1]Department of Meteorology, Stockholm University & Bolin Center for Climate Research, Stockholm, Sweden

[2]Laboratory of Atmospheric Processes and their Impacts, School of Architecture, Civil & Environmental Engineering, Ecole Polytechnique Fédérale de Lausanne, Lausanne, Switzerland

[3]Department of Space, Earth and Environment, , Chalmers University of Technology, Gothenburg, Sweden

[4]Institue for Chemical Engineering Sciences, Foundation for Research and Technology Hellas, Patras, Greece

Correspondence: georgia.sotiropoulou@misu.su.se

**Abstract**. Atmospheric models often fail to correctly reproduce the microphysical structure of Arctic mixed-phase clouds and underpredict ice water content, even when simulations are constrained by the observed levels of ice nucleating particles. In this study we investigate whether ice multiplication from

ice-ice collisions, a process missing in most models, can account for the observed cloud ice in a stratocumulus cloud observed during the Arctic Summer Cloud Study campaign. Our results indicate that including ice-ice collisions can improve the modeled cloud water properties, but the degree of influence depends on other poorly constrained microphysical aspects that include ice habit, rimed fraction and cloud ice-to-snow autoconversion rate. Simulations with dendrites are less sensitive to

variations in the assumed rimed fraction of the particle that undergoes break-up, compared to those with planar ice. Activating cloud ice-to-snow autoconversion decreases the sensitivity of the break-up process to both the assumed ice habit and rimed fraction. Finally, adapting a relatively small value for the threshold diameter at which cloud ice is converted to snow enhances break-up efficiency and improves the macrophysical representation of the cloud.

**Introduction**

Cloud feedbacks play an important role in Arctic climate change (Cronin and Tziperman, 2015; Kay et al., 2016; Tan and Storelvmo, 2019) and sea-ice formation (Burt et al., 2015; Cao et al., 2017).

However, despite their significant climatic impact, Arctic mixed-phase clouds remain a great source of uncertainty in climate models (Stocker et al., 2013; Taylor et al., 2019). To accurately predict the



radiative effects of mixed-phase clouds in models, an adequate description of their microphysical structure, such as the amount and distribution of both liquid water and ice, is required (Korolev et al., 2017). Both ice nucleation and liquid drop formation require seed particles to be present known as ice nucleating particles (INPs) and cloud condensation nuclei CCN), respectively. However, the observed ice crystal number concentrations (ICNCs) are often much higher than the observed INP concentrations in the Arctic (Fridlind et al., 2007; 2012; Gayet et al., 2009; Lloyd et al., 2015), where INPs are generally sparse (Wex et al., 2019). Moreover, model simulations constrained by INP measurements frequently underpredict the observed amount of ice (Fridlind and Ackerman, 2019).

Secondary Ice Processes (SIP) have been suggested as the reason why ice crystal concentrations exceed INP levels (Field et al., 2017; Fridlind and Ackerman, 2019). SIP involve the production of new ice crystals in the presence of pre-existing ice, without requiring the presence of an INP. The most well-known mechanism is rime-splintering (Hallet and Mossop, 1974), which refers to the ejection of ice splinters when ice particles collide with supercooled liquid drops. Rime-splintering is active only in a limited temperature range, between -8$^{o}$C and -3$^{o}$C, and requires the presence of liquid droplets both smaller than 13 μm and larger than 25 μm (Hallett and Mossop, 1974; Choularton et al., 1980). Moreover, recent studies have shown that rime-splintering alone cannot explain the observed ICNCs in polar clouds even within the optimal temperature range (Young et al., 2019; Sotiropoulou et al., 2020a,b). Ice fragments may also be generated when a relatively large drop freezes and shatters (Lauber et al., 2018; Phillips et al., 2018); drop-shattering, however, has been found insignificant in polar conditions (Fu et al., 2019; Sotiropoulou et al., 2020a). Finally, ice multiplication can occur from mechanical break-up due to ice-ice collisions (Vardiman et al., 1978; Takahashi et al., 1995). Despite that breakup has been observed in in-situ measurements of Arctic clouds (Rangno and Hobbs, 2001; Schwarzenboeck et al., 2009), it has received little attention from the modeling community.

Fridlind et al. (2007) and Fu et al. (2019) investigated the contribution from ice-ice collisions in an autumnal cloud case observed during the Mixed-Phase Arctic Cloud Experiment (M-PACE) and found that the process could not account for the observed ice content at in-cloud temperatures between -8.5$^{o}$C and -15.5$^{o}$C. The parameterization of the break-up process used in these studies was based on the laboratory data of Vardiman (1978). Phillips et al. (2017a,b) developed a more advanced treatment of ice multiplication from ice-ice collisions, which explicitly considers the ice collisional kinetic energy, ice habit, ice type and rimed fraction. Sotiropoulou et al. (2020a) and (2020b) applied this new formulation to a model study of polar clouds and concluded that it allowed ice enhancements that sufficiently explained the observed ICNCs. Both studies, however, focused on relatively warm polar clouds (-3$^{o}$C to -8$^{o}$C), where rime-splintering is also active.

In this study, we aim to investigate the role of ice-ice collisions at a somewhat colder in-cloud temperature range than in Sotiropoulou et al. (2020a) and (2020b). The simulated in-cloud range (~ -7$^{o}$C to –12.5$^{o}$C) includes temperatures for which previous parameterizations found limited efficiency of



the process (Fridlind et al., 2007; Fu et al., 2019). The Phillips parameterization is implemented in the MIT-MISU Cloud-Aerosol (MIMICA) Large Eddy Simulation (LES) modell to examine its
performance for a stratocumulus case study during the Arctic Summer Cloud Study (ASCOS) campaign in the high Arctic. To identify the optimal microphysical conditions for ice multiplication through collisional break-up, the sensitivity of the simulations to the assumed rimed fraction, ice habit and ice type (e.g. cloud ice/ snow) of the colliding ice particles is examined.

## 2. Field observations

The ASCOS campaign was deployed on the Swedish icebreaker *Oden* between 2 August and 9 September 2008 in the Arctic Ocean, to improve our understanding of the formation and life-cycle of Arctic clouds. It included an extensive suite of in-situ and remote sensing instruments, a description of which can be found in Tjernström et al. (2014). Here we only offer a brief description of the instruments
and measurements utilized in the present study.

### 2.1. Instrumentation

Information on the vertical atmospheric structure was derived from radiosondes, released every 6 hours. Cloud boundaries are derived from a vertically-pointing 35 GHz Doppler Millimeter Cloud
Radar (MMCR; Moran et al., 1998) and two laser ceilometers. CCN concentrations were measured by an in-situ CCN counter (Roberts and Nenes, 2005), set at a constant supersaturation of 0.2%, based on typical values used in other similar expeditions (Bigg and Leck, 2001; Leck et al., 2002). Vertically-integrated liquid water path (LWP) was retrieved from a dual-channel microwave radiometer, with an uncertainty of 25 g m$^{-2}$ (Westwater et al., 2001). Ice water content (IWC) was estimated from the radar
reflectivity observed by the MMCR, using a power-law relationship (e.g. Shupe et al, 2005), with a factor of 2 uncertainty; The ice water path (IWP) was integrated from the IWC estimates.

### 2.2. ASCOS case study

A detailed description of the conditions encountered during the ASCOS campaign is available in
Tjernström et al. (2012). Our focus here is on a stratocumulus deck observed between 30-31 August, while *Oden* was drifting with a 3×6 km$^2$ ice-floe at approximately 87° N. During that time, relatively quiescent large-scale conditions prevailed, characterized by a high-pressure system and large-scale subsidence in the free troposphere and only weak frontal passages (Tjernström et al., 2012).

Our simulations are initialized with thermodynamic and cloud liquid profiles representing
conditions observed on 30 August at 18 UTC (Fig. 1). These profiles display a cloud layer between 550 and 900 m above ground level (a.g.l), at temperatures between -7°C and -10°C, capped by a temperature and humidity inversion, of about 5°C and 0.5 g kg$^{-1}$, respectively. A weak secondary temperature inversion is also observed at about 370 m a.g.l., indicating that the cloud is decoupled from the surface;





this type of vertical structure, with a decoupled surface and cloud layer, dominated during the whole

ASCOS experiment (Sotiropoulou et al., 2014). More generally, this case study is representative of typical cloudy boundary layers over sea-ice, where co-existing temperature and humidity inversions are frequently observed (Sedlar et al., 2012), and clouds are often decoupled from any surface sources of e.g. moisture (Sotiropoulou et al., 2014).

The observed cloud layer remained 'stable' for about 12 hours from the selected starting time

and began dissipating after 31 August 9 UTC. A substantial reduction in the background aerosol concentration has been suggested as a possible cause for the sudden collapse of the cloud layer, which cannot be simulated by models without prognostic aerosol processes (Stevens et al., 2018). For this reason, we will use observational statistics from this period with the persistent stratocumulus conditions to evaluate our results, although simulations are allowed to run for 24 hours in a quasi-equilibrium state.


## 3. Model and Methods

### 3.1. LES set-up

The MIMICA LES (Savre et al., 2014) solves a set of non-hydrostatic prognostic equations for the conservation of momentum, ice-liquid potential temperature and total water mixing ratio with an

anelastic approximation. A fourth order central finite-differences formulation determines momentum advection and a second order flux-limited version of the Lax-Wendroff scheme (Durran, 2010) is employed for scalar advection. Equations are integrated forward in time using a second order Leapfrog method and a modified Asselin filter (Williams, 2010). Subgrid scale turbulence is parameterized using the Smagorinsky-Lilly eddy-diffusivity closure (Lilly, 1992) and surface fluxes are calculated according

to Monin-Obukhov similarity theory.

Cloud microphysics is described using a two-moment approach for cloud droplets, rain and cloud ice, graupel and snow particles. Mass mixing ratios and number concentrations are treated prognostically for these five hydrometeor classes, whereas their size distributions are defined by generalized Gamma functions. Cloud droplet and raindrop processes follow Seifert and Beheng (2001),

while liquid/ice interactions are parameterized as in Wang and Chang (1993). A simple parameterization for CCN activation is applied (Khvorostyanov and Curry, 2006), where the number of cloud droplets formed is a function of the modeled supersaturation and a prescribed background aerosol concentration ($N_{CCN}$). A detailed radiation solver (Fu and Liou, 1992) is coupled to MIMICA to account for cloud radiative properties when calculating the radiative fluxes.

The model configuration adopted is based on Ickes et al. (in prep.), who simulated the same case to examine the performance of various primary ice nucleation schemes. All simulations are performed on a 96×96×128 grid, with constant horizontal spacing d$x$ = d$y$ = 62.5 m. The simulated domain is 6×6 km$^2$ horizontally and 1.7 km vertically. At the surface and in the cloud layer the vertical grid spacing is 7.5 m, while between the surface and the cloud base it changes sinusoidally, reaching a maximum





spacing of 25 m. The integration time step is variable (~ 1-3 sec), calculated continuously to satisfy the
Courant-Friedrichs-Lewy criterion for the Leapfrog method. While this approach prevents numerical
instabilities, its dynamic nature does not allow sensitivity simulations to be performed with exactly the
same timestep. Lateral boundary conditions are periodic, while a sponge layer in the top 400 m of the
domain dampens vertically-propagating gravity waves generated during the simulations. To accelerate

the development of turbulent motions, the initial ice-liquid potential temperature profiles are randomly
perturbed in the first 20 vertical grid levels with an amplitude less than $3\times10^{-4}$ K.

Surface pressure and temperature are set to 1026.3 hPa and -3.2°C, respectively, constrained by
surface sensors deployed on the ice-pack. The surface moisture is set to the saturation value, which
reflects summer ice conditions. The surface albedo is assumed to be 0.85, which is representative of a

multi-year ice pack. In MIMICA, subsidence is treated as a linear function of height: $w_{LS} = -D_{LS} \cdot z$,
where $D_{LS}$ is set $1.5\times10^{-6}$ s$^{-1}$ and $z$ is the height in meters. Finally, the prescribed number of CCN is set
to 30 cm$^{-3}$ over the whole domain, which represents mean accumulation mode aerosol concentrations
observed during the stratocumulus period (cf. Igel et al., 2017). The duration of all simulations is 24
hours, where the first 4 hours constitute the spin-up period.


### 3.2 Ice Formation Processes in MIMICA

#### 3.2.1 Primary ice production

Ickes et al. (in prep.) recently implemented several primary ice production (PIP) schemes in

MIMICA. Here, we utilize the empirical ice nucleation active site density parameterization for
immersion freezing, which is based on Connolly et al. (2009) and was further developed by Niemand et
al. (2012) for Saharan dust particles. This formulation was used by Ickes et al. (2017) to describe the
freezing behavior of different dust particle types, including microline (Appendix A). Microcline is a
feldspar type that is known to be an efficient INP (Atkison et al., 2013). As no aerosol composition (or

INP) measurements are available for the ASCOS campaign, we will use this INP type as a proxy for an
aerosol constituent that can produce primary ice at the relatively warm sub-zero temperatures (-7°C to -
10°C) of the initial observed cloud profile (Fig. 1a). At these temperatures it is reasonable to assume
that most of the PIP occurs through immersion freezing (Andronache, 2017), i.e. that an aerosol must be
both CCN active and contain ice-nucleating material to initiate ice production. Thus, we will simply

assume that a specified fraction of the CCN population contains some efficient ice-nucleating material,
here represented as feldspar, and match this fraction so that the model simulates reasonable values of
LWP, IWP and ICNC (Appendix A, Text S1). Based on this procedure, we specify that the CCN
population contains 5% microline, a value that results in realistic primary ICNCs (Wex et al., 2019), but
in an underestimate of the IWP and an overestimate of the LWP (Text S1, Fig. S1). Note that even

though we assume this relatively high fraction of ice-nucleating material (Text S1, Fig. S1), MIMICA





still underestimates the IWP; we hypothetize that secondary ice production may be the reason for this bias.

### 3.2.2 Ice multiplication from ice-ice collisions

The observed in-cloud temperatures are generally below the rime-splintering temperature range, except for the somewhat warmer temperatures near cloud base (Fig. 1a). Moreover, drop-shattering has been found ineffective for Arctic conditions (Fu et al., 2019; Sotiropoulou et al., 2020a). Our simulations further support the inefficiency of both these processes, as the concentration of large raindrops is too low (below 0.1 cm$^{-3}$) to initiate them (Fig. S1c). Hence we focus solely on ice

multiplication from ice-ice collisions.

We implement the parameterization developed by Phillips et al. (2017a) in MIMICA and allow for ice multiplication from cloud ice-cloud ice, cloud ice-graupel, cloud ice-snow, snow-graupel, snow-snow and graupel-graupel collisions (Appendix B). The generated fragments are considered "small ice" crystals and are added to the cloud ice category in the model. The Phillips parameterization explicitly

considers the effect of ice type, ice habit and rimed fractions of the colliding particles on fragment generation (Appendix B). The sensitivity of the model performance to these parameters is examined through sensitivity simulations.

### 3.3 Sensitivity simulations

A detailed description of the sensitivity tests is provided in this section, while a summary is offered in Table 2.

### 3.3.1 The role of ice habit

Cloud ice observed within the examined temperature range can either be shaped as a dendrite or

a plate, depending on the supersaturation with respect to ice (Pruppacher and Klett, 1997). However, as the mean vapor density excess in the simulated cloud layer varies between 0.03 and 0.22 g m$^{-3}$, it is not clear which shape should theoretically dominate (Pruppacher and Klett, 1997). Moreover, observations often indicate variable shapes within the same temperature conditions (Mioche et al., 2017). The formulations for ice multiplication due to break-up are substantially different for these two ice habits

(Appendix B). Generally, the number of fragments ($F_{BR}$) generated from break-up of a dendrite is lower than that generated by a plate, assuming that the two particles have the same mass and size (Appendix B).

MIMICA allows for variable treatment of the ice habit for the cloud ice category. These variations correspond to different characteristic parameters in the $(m = a_m D^{b_m})$ and fallspeed-diameter

$(v = a_v D^{b_v})$ relationships, whose values (Table 1) are adopted from Pruppacher and Klett (1997) and Khvorostyanov and Curry (2002). To test the sensitivity of our results to the assumed cloud ice habit,





the two simulations CNTRLDEN and CNTRLPLA are performed. 'CNTRL' refers to simulations that account only for PIP, while the suffixes 'DEN' and 'PLA' indicate dendritic and planar cloud ice shape, respectively.


### 3.3.2 The role of rimed fraction

$F_{BR}$ is parameterized as a function of the rimed fraction ($\Psi$) of the ice crystal or snowflake that undergoes break-up; fragment generation from break-up of graupel does not depend on $\Psi$ (see Appendix B). This parameter is not explicitly predicted in most bulk microphysics schemes, but can substantially affect the multiplication efficiency of the break-up process (Sotiropoulou et al., 2020a). For this reason, we will consider values of $\Psi$ for cloud ice and snow between 0.2 (lightly rimed) and 0.4 (heavily rimed) (Phillips et al., 2017a, b); graupel particles are considered to have $\Psi \geq 0.5$. Both Sotiropoulou et al. (2020a) and (2020b) found that ice multiplication in polar clouds at temperatures above -8$^o$C is initiated only when a highly rimed fraction of cloud ice and snow is assumed. Their conclusions however may not be valid for our case, as the temperature and microphysical conditions are substantially different.

The effect of varying $\Psi$ is examined for the two ice habits that prevail in the observed temperature range (Section 3.2.2). The performed simulations are referred to as BRDEN0.2, BRDEN0.3, BRDEN0.4, and BRPLA0.2, BRPLA0.3, BRPLA0.4, for dendrites and plates respectively (see Table 2). 'BR' indicates that collisional break-up is active, while the number 0.2-0.4 corresponds to the assumed value of $\Psi$.

### 3.3.3 The impact of the representation of the ice particle spectrum

The MIMICA LES has previously been used to study ice-ice collisions in Sotiropoulou et al. (2020a), however, they used a parcel-model-based parameterization of the process, instead of implementing a break-up parameterization as a part of the MIMICA microphysics scheme. Sotiropoulou et al. argued that the efficiency of the process is likely underestimated in bulk microphysics schemes, where the dynamics of the ice particle spectrum is poorly represented and fixed particle properties are assumed typically for three ice types (cloud ice, graupel, snow), which is rather unrealistic. Their argument might be particularly true for the studied case where no snow is produced in the simulations with dendrites (Fig. S1d). The MIMICA microphysics scheme allows for snow formation only through aggregation of cloud ice particles. However, these are particularly few in the CNTRLDEN simulation and thus collisions between them are negligible. Other microphysics schemes (Morrison et al., 2005; Morrison and Gettelman, 2008) typically include an ice-to-snow autoconversion parameterization, which means that cloud ice particles become snowflakes once their size exceeds a specified critical diameter ($D_c$).

To test the impact of a broader ice particle spectrum on the multiplication effect of break-up, we implement a description for ice-to-snow autoconversion in MIMICA (Appendix C). Eidhammer et al.





(2014) showed that cloud ice properties can be sensitive to the assumed separation (or critical) diameter ($D_c$); this parameter usually varies between 100-500 µm in existing microphysics schemes (e.g.
Morrison et al. 2005; Morrison and Gettelman, 2008; Hong et al., 2004). Here we test two values: (a) $D_c$ =125 µm, adapted from Morrison et al. (2005), and a larger value (b) $D_c$ =500 µm (Hong et al., 2004). The simulations with an active autoconversion scheme are referred to as: (a) CNTRLDENauto1, BRDEN0.2auto1 and BRDEN0.4auto1 when $D_c$ =125 µm and (b) CNTRLDENauto2, BRDEN0.2auto2 and BRDEN0.4auto2 when $D_c$ =500 µm. As explained above, 'CNTRL' indicates the simulation that
accounts only for PIP, while collisional break-up (BR) is tested for two assumed rimed fractions: 0.2 and 0.4. The ice-to-snow autoconversion scheme is also activated in simulations with planar ice, even though these actually allow for snow formation through aggregation (see section 4.1). These tests are referred as: (a) CNTRLPLAauto1, BRPLA0.2auto1, BRPLA0.4auto1 and (b) CNTRLPLAauto2, BRPLA0.2auto2, BRPLA0.4auto2 (see Table 2).


## 4. Results

### 4.1 Sensitivity to ice habit

The impact of the assumed ice habit is investigated using the two simulations that account only
for PIP (CNTRLDEN and CNTRLPLA) and two of the simulations that include break-up: BRDEN0.2 and BRPLA0.2. Time series of LWP and IWP are shown in Fig. 2, while the domain-averaged ice particle concentrations for the three ice categories are shown in Fig. 3. Median and interquartile statistics are summarized in Table 3, while the ICNC enhancement from break-up is shown in the Supplementary Information (Text S2, Fig. S2).

Small differences are observed in the integrated cloud water quantities between CNTRLDEN and CNTRLPLA, as both produce median LWP values ~143 g m$^{-2}$ and median IWP values of 1.8-1.9 g m$^{-2}$. Hence, both simulations overestimate cloud liquid (Fig. 3a-b) and underestimate ice (Fig. 2c-d). Specifically, the median observed LWP (73.8 g m$^{-2}$) is overestimated by almost a factor of two, while IWP (7 g m$^{-2}$) is underestimated by more than a factor of 3.5 (Table 3), which is larger than the
uncertainty in the observations. Activating break-up results in improved simulated water properties, independently of the assumed ice habit. LWP decreases by 32 g m$^{-2}$ (37 g m$^{-2}$) in BRDEN0.2 (BRPLA0.2) compared to CNTRLDEN (CNTRLPLA) at the end of the simulation. However, in both BR simulations, the LWP remains above the observed interquartile range (Fig. 2a-b): the median LWP is 105 and 119 g m$^{-2}$ in BRDEN0.2 and BRPLA0.2. The simulated IWP is also improved when break-
up is considered, as it remains within the observed interquartile range for most of the simulation time (Fig. 3d). Statistical metrics of IWP are shifted to larger values in BRDEN0.2 compared to BRPLA0.2 (Table 3), suggesting a better agreement of BRDEN0.2 with observations.


Planar ice is expected to generate more fragments per collision compared to plates if the diameter of the particles and the collisional kinetic energy are the same (see equations 6-7 in Appendix B). However, BRDEN0.2 eventually produces about the same or slightly more ice compared to BRPLA0.2 (Fig. 2c-d). Figure 4a shows that the cloud ice number ($Ni$) enhancement due to break-up of dendrites is more consistent compared to planar ice (Fig. 3b, Table 3). $Ni$ for planar ice displays large fluctuations; large $Ni$ enhancements are followed by substantial decreases (Fig. 3b). This variability indicates that precipitation processes (i.e. the precipitation sink) are more effective in the simulations with plates, which is linked to the larger characteristic mass and terminal velocity specified for this ice habit, compared to dendrites of similar size (Table 1). The larger lifetime of dendritic $Ni$ in BRDEN0.2 within the cloud layer leads to more frequent collisions with liquid drops and thus enhanced graupel formation (Fig. 3c), compared to BRPLA0.2 (Fig. 3d). Fragment generation in BRDEN0.2 further enhances cloud ice aggregation compared to CNTRLDEN as indicated by the fact that snow formation is only favored in the former (Fig. 3e). This is not always the case for BRPLA0.2 where enhanced removal of $Ni$, as discussed above, suppresses ice-ice collisions and thus snow formation compared to CNTRLPLA (Fig. 3f).

Despite the different feedbacks between ice multiplication and precipitation in BRDEN0.2 and BRPLA0.2, ICNC enhancement rarely exceeds a factor of 2 when compared to the CNTRLDEN and CNTRLPLA simulations (Text S2, Fig. S2). However, this weak enhancement found for the ASCOS case can still bridge the gap between observed and modeled ice water content, especially when a dendritic ice habit is assumed. The median IWP enhancement is approximately 2.7 and 3.3 in the BR simulations with dendrites and plates, respectively. Moreover, including break-up reduces the bias in LWP, although this parameter remains overestimated by the model, independently of the chosen ice habit. Stevens et al. (2018) showed that simulations with interactive aerosols produce less LWP during the examined case, compared to simulations with a fixed background CCN concentration. Thus deviations between the simulated and observed LWP could also be attributed to the simplified aerosol treatment, rather than to uncertainties in the representation of the break-up process.

## 4.2. Sensitivity to rimed fraction

Since the rimed fraction of cloud ice/snow particles is not explicitly predicted in MIMICA, we investigate the sensitivity of the simulated LWP and IWP to the prescribed $\Psi$ value (Fig. 4). To better understand the differences between the simulations, fragment generation rates ($P_{BR}$) for the different collision types are shown in Fig. 5 (see Appendix B for detailed formulas). $P_{BR}$ results are only presented for cloud ice-graupel, graupel-snow and snow-snow collisions since we find negligible contributions from cloud ice-cloud ice, cloud ice-snow and graupel-graupel collisions.

All sensitivity simulations with dendrites produce similar LWP values (Fig. 4a), constantly remaining above the observed interquartile range. Small differences are also found in IWP (Fig. 4c).



The median LWP fluctuates between 104.8-110.9 g m$^{-2}$ and IWP between 5-6.3 g m$^{-2}$ among the three
sensitivity simulations (Table 3), suggesting that cloud liquid remains overestimated while ice
properties are in agreement with the observed range. The total fragment generation from all collisions
remains constantly below 1.4 L$^{-1}$s$^{-1}$ with small differences between the three simulations with varying $\Psi$
(Fig. 5a,c,e). Fragments are mainly generated by graupel-snow collisions in BRDEN0.2 and BRDEN0.3
(Fig. 5a,c), while snow-snow collisions become important in BRDEN0.4. Nevertheless, the total $P_{BR}$
remains very similar in all these simulations (Fig. 5e), while ICNC enhancement is on average about a
factor of 2 (Text S2, Fig. S2).

The effect of $\Psi$ is more evident in the simulations with planar shape (Fig. 4b,d). BRPLA0.2 is
the only simulation that sustains a cloud layer throughout the simulated period (Fig. 4b), since total $P_{BR}$
remains below 2.2 L$^{-1}$s$^{-1}$ throughout the simulation time (Fig. 5b,d,f) and ICNC enhancement never
exceeds a factor of 3 (Fig. S2). The cloud dissipates in BRPLA0.3 and BRPLA0.4 after 17 and 7 hours,
respectively (Fig. 4b). The cloud in BRPLA0.4 rapidly dissipates owing to excessive multiplication,
with the total $P_{BR}$ reaching a maximum of 73.6 L$^{-1}$s$^{-1}$ (Fig. 5b,d,f). However, the cloud reforms after 15
hours; LWP increases again to values larger than 100 g m$^{-2}$ by the end of the simulated period (Fig. 4b),
while IWP remains at substantially underestimated levels compared to observations (Fig. 4d). The total
fragment generation is more moderate in BRPLA0.3 than in BRPLA0.4 reaching a maximum rate of
14.6 L$^{-1}$s$^{-1}$ after 14.5 hours (Fig. 5b,d,f), causing a gradual cloud glaciation. ICNCs in BRPLA0.3 are
enhanced by up to a factor of 80, compared to CNTRLPLA, while in BRPLA0.4 the ICNC
enhancement can be up to three orders of magnitude (Fig. S2). Our findings are in agreement with
Loewe et al. (2018) who showed a prescribed ICNC value of 10 L$^{-1}$ can lead to cloud dissipation for the
specific case study. However, collisional break-up of moderately to highly rimed plates is likely not the
reason that lead to cloud dissipation after 31 August 9 UTC in reality (see Section 2.2). In the next
section we will show that the excessive multiplication in these two simulations is due to limited snow
formation in this model set-up.

Variations in ice habit do not result in significant changes in cloud properties when a low $\Psi$
(~0.2) is assumed (see Section 4.1). However, the assumed ice habit significantly impacts cloud
conditions for moderate or high $\Psi$ values. Furthermore, in the simulations with dendrites, statistics of
cloud water properties show low sensitivity to variations in $\Psi$ (Table 3). In contrast, errors in $\Psi$ could
substantially change the results for planar ice, as it exerts significant control on cloud life-cycle.

**4.3 Sensitivity to the representation of the ice particle spectrum**

Implementing cloud ice-to-snow autoconversion in MIMICA is expected to alter the
representation of the ice particle spectrum and thus impact the efficiency of the break-up process (see
Section 3.3.3). The simulation with a relatively low critical diameter for autoconversion, set to 125 μm,



is examined in subsection 4.3.1. The simulations with the cloud ice/snow separation diameter set to 500
μm is discussed in subsection 4.3.2.

### 4.3.1 Cloud ice-to-snow autoconversion with critical diameter 125 μm

CNTRLDENauto1 produces somewhat larger IWP than CNTRLDEN (Fig. 6c) and falls within
the observed range after 18 hours of simulation, however, LWP values remain overestimated (Fig. 6a).
These small increases (decreases) in IWP (LWP) in simulations with autoconversion are associated with
larger ICNCs; the maximum concentration is about two times larger in CNTRLDENauto1 (Fig. S3)
compared to CNTRLDEN (Fig. S2). This is due to the fact that conversion of cloud ice to snow
increases snow formation but at the same time results in fewer graupel particles (not shown), likely due
to decreasing efficiency of cloud-ice and liquid droplet collisions. As graupel particles have larger
terminal velocities than snow (see Section 3.3.1), the net result of decreasing (increasing) graupel
(snow) formation is less precipitation and thus enhanced ICNCs within the cloud layer. Nevertheless,
limiting the ice precipitation sink in CNTRLDENauto1 still results in underestimated (overestimated)
IWP (LWP) statistics (Table 3); the median IWP and LWP is 4 g m$^{-2}$ and 130 g m$^{-2}$, respectively.
In simulations with break-up, BRDEN0.2auto1 and BRDEN0.4auto1, IWP increases in better
agreement with reality (Fig. 6b, Table 3). While median LWP remains overestimated compared to
observations (Table 3), these are the only simulations so far that result in LWP values (Fig. 6a) that
eventually match the observed range (at least within the last 9 hours of simulation). This result is more
clearly seen in Fig. 7: while CNTRLDEN and CNTRDENauto1 fail to reproduce the observed relative
frequency of the LWP-IWP fields, the frequency distributions of BRDEN0.2auto1 and BRDEN0.4auto1
are shifted towards more realistic values. Also note that while BRDEN0.4auto1 produces statistical
metrics closer to the observations (Table 3),  BRDEN0.2auto1 gives a better representation of the
LWP/IWP variability (Fig. 7).
Including snow autoconversion in simulations with planar ice that account only for PIP also
results in enhanced ICNCs (Fig. S3) due to decreasing graupel formation and thus decreasing
precipitation (see discussion above). This explains the IWP enhancement in CNTRLPLAauto1
compared to CNTRLPLA (Fig 6d, Table 3). Yet, median IWP remains underestimated more than a
factor of two in this simulation (3.1 g m$^{-2}$) compared to observations (7 g m$^{-2}$), while LWP statistics are
substantially overestimated (Table 3). Moreover, autoconversion does not improve substantially the
simulated LWP-IWP relationship compared to ASCOS measurements (Fig. 8). Activating break-up for
this set-up improves modeled cloud ice properties, as BRPLA0.2auto1 and BRPLA0.4auto1 produce
median IWP at 6.1 g m$^{-2}$ and 6.4 g m$^{-2}$. The LWP bias is also improved, with median values 89.6 g m$^{-2}$
and 77.8 g m$^{-2}$ in BRPLA0.2auto1 and BRPLA0.4auto1, respectively. Furthermore, these simulations
produce more realistic distributions of the LWP-IWP fields compared to set-ups that do not account for





break-up (Fig. 8). From all simulations presented so far, BRPLA0.4auto1 is the one that produces the most realistic interquartile range for cloud liquid properties (Table 3).

### 4.3.2 Cloud ice-to-snow autoconversion with critical diameter 500 μm

Adapting a larger cloud ice/snow separation diameter results in enhanced (reduced) liquid (ice)
water properties (Fig. 9). This is because lower cloud ice concentrations are converted to snow, which enhances graupel formation (not shown) through cloud ice - droplet collisions and thus ice precipitation (see discussion above). In particular, ICNCs in CNTRLDENauto2 and CNTRPPLAauto2 do no exceed 1.3 $L^{-1}$ (Fig. S4), which is about 38% smaller than the maximum values found in CNTRLDENauto1 and CNTRLPLAauto1 (Fig. S3). Moreover, water paths in these simulations are very similar to
CNTRLDEN and CNTRLPLA (Fig. 9, Table 3).

Activating break-up for this set-up results in median IWP=5.4 g $m^{-2}$ in both BRDEN0.2auto2 and BRDEN0.4auto2, while the median LWP values are 103.8 and 106.9 g $m^{-2}$, respectively. These are lower (larger) than the median IWP (LWP) results produced in the corresponding runs with $D_c$=125 μm (Table 3). In particular, BRDEN0.2auto2 and BRDEN0.4auto2 produce more similar results as the
simulations that do not account for autoconversion (BRDEN0.2 and BRDEN0.4 in Fig. 2 and Table 3).

Simulations with plates respond in a similar manner to variations in separation diameter as those with dendrites (Fig. 9b,d). CNTRLPLAauto2, BRPLA0.4auto2 and BRPLA0.2auto produce increased (reduced) liquid (ice) water properties compared to CNTRLPLAauto1, BRPLA0.2auto1 and BRPLA0.4auto1, respectively (Table 3). However, while CNTRLPLAauto2 and BRPLA0.2auto2
produce similar results as the corresponding simulations with inactive autocoversion (CNTRLPLA, BRPLA0.2), this is not the case for BRPLA0.4auto2. While BRPLA0.4 results in cloud glaciation, BRPLA0.4auto2 sustains the cloud layer throughout the simulation. This indicates that activating cloud ice-to-snow autoconversion can moderate secondary ice production and prevent cloud dissipation, independently of the assumptions in separation diameter.

The large differences in the cloud life-cycle produced by BRPLA0.4 and BRPLA0.4auto1/BRPLA0.4auto2 occur because small ice fragments accumulate within the cloud layer in the former simulation (Fig. 6b), continuously feeding the break-up process. In contrast, active autoconversion eventually converts the new ice fragments into snow, allowing for their faster depletion through precipitation. In other words, as precipitation processes are more effective for snow than for
cloud ice, snow formation prevents accumulation of cloud ice particles and thus decreases the frequency of ice particle collisions (Figs 5b,d,f and 10b,d,f). This snow precipitation feedback inhibits the explosive multiplication (Yano and Phillips, 2011; 2016) observed in the simulations that do not allow for cloud ice depletion through the autoconversion process.

The better agreement with observed LWP and IWP for the simulations with a separation
diameter of 125 μm compared to those with 500 μm is due to changes in a number of ice processes that



impact each other. Increasing graupel formation with increasing $D_c$ (see discussion above) results in more frequent snow-graupel collisions; $P_{BRsg}$ is on average about 3 to 5 times larger in simulations with $D_c$ = 500 μm compared to those with a similar set-up but $D_c$ = 125 μm (Fig. 10c-d). For the same reason, cloud ice-graupel collisions also become more frequent with increasing $D_c$, but the fragment

generation rate of this collision type is generally small (Fig. 10a-b). Meanwhile, the increased snow formation through decreasing $D_c$ results in more efficient snow-snow collisions (Fig. 10e-f). This collision type results in larger fragmentation rates when $D_c$ is set to 125 μm, eventually exceeding 1 L$^{-1}$s$^{-1}$ (2 L$^{-1}$s$^{-1}$) in simulations with plates (dendrites); these are on average about 7-8 (8.5-10) times larger than the corresponding rates in simulations with $D_c$ = 500 μm. Thus while changes in $D_c$ impact all

collision types, the changes in snow-snow collision efficiency are mainly responsible for the differentiations in cloud water properties observed between the two simulation set-ups (Figs. 6, 9).

      In summary, while it was concluded in the previous subsection that the assumed ice habit for highly-rimed ice particles can result in completely different cloud conditions (Fig. 4), this sensitivity is substantially smaller when ice-to-snow autoconversion is active (Fig. 6, 9). At the same time, while the

assumed $\Psi$ played a significant role in the simulations with plates presented in Fig. 4b, its impact is substantially decreased when autoconversion is allowed (Fig. 6,9b). Finally, the mean ICNC enhancement due to break-up in all these simulations remains weak, rarely exceeding a factor of 2-3 (Fig. S3-4). Yet, despite the weak ice enhancement, activation of break-up can still substantially impact the simulated macrophysical properties of the cloud (Figs. 7-8), provided that cloud ice-to-snow

autoconversion at a relatively low $D_c$ is adopted.

## 5. Discussion

      Ice formation processes in Arctic clouds are sources of great uncertainty in atmospheric models, often resulting in underestimation of the cloud ice content. In this study we attempt to quantify the

impact of ice multiplication through collisional break-up for summertime high-Arctic conditions and examine the sensitivity of the efficiency of this process to a number of uncertain parameters, such as the ice habit, ice type and rimed fraction of the colliding particles.

      Uncertainties in ice habit are in general not important as long as a low rimed fraction (~0.2) is assumed. However, while changes in the rimed fraction have a small impact on the simulations with

dendrites, break-up of moderately to highly rimed particles can result in explosive multiplication in the simulations with plates, if cloud ice-to-snow autoconversion is not accounted for in the model. This is because the freshly formed small fragments can stay long and accumulate in the cloud ice category, continuously feeding the multiplication process as they grow, until the cloud glaciates. If a substantial amount of the new planar fragments would be converted to snow, then this ice type has a shorter

lifetime within the cloud layer. As a result, the continuous feeding of break-up is balanced by precipitation processes, which prevents cloud dissipation.



Activation of cloud ice-to-snow autoconversion generally decreases the sensitivity of the simulation results to both the assumed ice habit and the rimed fraction. This result indicates that if the autoconversion process allows for sufficient snow formation, there is no great need to accurately

constrain these two parameters in the description of break-up; note that there are limited observations of rimed fraction and ice habits in the Arctic, which are highly variable parameters (Mioche et al., 2017). The break-up efficiency can be largely affected by the separation diameter prescribed in the autconversion process; this is the threshold diameter at which cloud ice is converted to snow. This parameter has little physical basis (Eidhammer et al., 2014) and cannot be measured directly, thus it is

generally tuned in atmospheric models.

Our simulations suggest that decreasing the separation diameter for cloud ice and snow promotes snow formation and thus snow-snow collisions, which is a very efficient collision type. As a result, adapting a relatively small threshold improves the representation of cloud water properties and especially cloud liquid content, which is substantially overestimated in all other simulation set-ups.

However apart from ice-to-snow autoconversion, other factors can also contribute to decreasing liquid properties, such as the prognostic treatment of aerosols (Stevens et al., 2018). Thus, while with the current model setup with fixed background aerosol concentrations, a low separation diameter results in more realistic cloud properties, it is uncertain whether the same tuning would work for other models with a more advanced aerosol treatment.

Nevertheless, ICNC enhancement in the most realistic simulations rarely exceeds a factor of 2-3. This suggests a lower efficiency of the break-up process in the examined temperature range, compared to warmer polar conditions studied by Sotiropoulou et al. (2020a,b). In their studies they focused on the Hallet-Mossop temperature range characterized by lower INP concentrations that do not exceed 0.1 $L^{-1}$ and found a 10-20 fold enhancement in ICNCs due to break-up compared to the available INPs.

However in sensitivity tests of primary ice nucleation, they showed that increasing INPs result in decreasing secondary ice production.

In the present study, relatively high INP conditions are adapted. Primary ICNCs increase with time as cloud cools through radiative cooling, reaching a maximum of 1 $L^{-1}$ towards the end of the simulation; these concentrations are even larger in simulations with active cloud ice-to-snow

autoconversion. While primary ice formation in our set-up is likely overestimated (Fridlind et al., 2007; Wex et al., 2019), our results support the conclusions of Sotiropoulou et al. (2020a,b) and further suggest that as primary ice nucleation becomes more and more enhanced at colder temperatures, ice multiplication from ice-ice collisions will likely become less significant. It is interesting that while laboratory experiments from Takahashi et al. (1995), based on collisions of two hailstones, suggest

increasing ice multiplication with decreasing temperature from -3$^{o}$C to -15$^{o}$C, our findings indicate that this might not happen in the real atmosphere due to increasing availability of INPs.





## 6. Conclusions

In this study, ice multiplication from ice-ice collisions is implemented in the MIMICA LES,
following Phillips et al. (2017a,b), to investigate the role of this process for ice-liquid partitioning in a
summertime Arctic low-level cloud deck observed during ASCOS. The sensitivity of the simulated
results to the prescribed ice habit, rimed fraction and the ice particle spectrum of the colliding particles
(i.e. by including ice-to-snow autoconversion) is investigated. Our findings can be summarized as
follows:


- For the simulated temperature range (-12.5 to -7 $^{o}$C), ice multiplication from collisional break-up is
  generally weak, enhancing ICNCs by on average no more than a factor of 2-3 in simulations that are
  most consistent with observations. This enhancement can result in a 2-3 fold increase in median
  IWP and deplete median LWP by 30-40%, compared to simulations that do not account for this
process. Simulation set-ups that produce very large ICNC enhancements, up to 2-3 orders of
  magnitude, result in cloud glaciation.
- Ice multiplication from break-up of dendrites is not very sensitive to assumptions regarding the
  rimed fraction. Break-up of lightly rimed planar ice also results in similar cloud water as simulations
  with dendrites. In contrast, break-up of highly rimed plates can lead to cloud glaciation, if cloud ice-
to-snow autoconversion is not accounted for in the microphysics scheme.
- Activating cloud ice-to-snow autoconversion results in substantially reduced sensitivity to the
  assumed ice habit as well as the rimed faction. Decreasing the critical diameter in the
  autoconvertion process, and thus allowing for more snow to form, enhances break-up efficiency.
  This model set-up substantially improves the simulated distribution of the LWP-IWP fields
compared to observations.

While ice enhancement due to ice-ice collisions in Arctic clouds is weaker within the examined
temperature range compared to warmer sub-zero temperatures and lower INP conditions (Sotiropoulou
et al. 2020a,b), including this process in models can still have a significant impact on the cloud
macrophysical state. However, processes that govern the ice particle spectral representation, such as
cloud ice-to-snow autoconversion in bulk microphysics schemes, substantially affect the simulated
collisional break-up. In particular, uncertainties in the separation diameter between cloud ice and snow
can have a larger effect on the break-up efficiency than poorly constrained parameters that are directly
included in the description of the process (i.e. ice habit or rimed fraction). Hence, to capture the effects
of ice multiplication from ice-ice collisions, it is important to better understand the cloud ice-to-snow
autoconversion process and to evaluate the separation parameter through model-observation
comparisons for the range of atmospherically-relevant conditions.



**Code and data availability:** ASCOS data are available at https://bolin.su.se/data/ascos/ . The modified
LES code is available upon request

**Competing interests:** The authors declare that they have no conflict of interest.

**Author contribution:** GS implemented the break-up parameterization in the LES, performed the
simulations, analyzed the results and led manuscript writing. LI implemented the primary ice nucleation
scheme. All authors contributed to the scientific interpretation, discussion and writing of the
manuscript.

**Acknowledgements:** The authors acknowledge support from the project IC-IRIM (project ID 2018-
01760) funded by the Swedish Research Council for Sustainable Development (FORMAS) and the
project FORCeS funded from Horizon H2020-EU.3.5.1. (project ID 821205). They are also grateful to
ASCOS scientific crew for the observational datasets used in this study. The computations were enabled
by resources provided by the Swedish National Infrastructure for Computing (SNIC) at the National
Supercomputer Centre (NSC) partially funded by the Swedish Research Council through grant
agreement no. 2016-07213.

**APPENDIX A: PRIMARY ICE PRODUCTION**

The immersion freezing parameterization is based on the concept of ice nucleation active site density.
The formulation of Niemand et al. (2012) is used, adapted for microline dust particles (Ickes et al.,
2017). It is utilized here as the only primary ice production mechanism. In this scheme, the number of
nucleated ice particles ($N_{INP}$, m$^{-3}$) is given as function of $N_{CCN}$ and temperature $T$ ($^o$C) :

$$N_{INP} \ = XN_{CCN}\left(1 - e^{-4\pi r^2 n_s}\right) \ ,$$

where $n_s = e^{-aT+b}$. $X$ is the percentage of $N_{CCN}$ (m$^{-3}$) that acts as efficient INP, e.g. 50%, 10%, 5%
(see Text S1 in Supporting Information) and $n_s$ (m$^{-2}$) the ice nucleation active site density of the INP
species assumed (here microline). $r$ =46.5$^.$10$^{-9}$ m is the mean radius of the nucleating particles for
ASCOS (Ickes et al., in prep.). The temperature dependency is determined by the coefficients
$\alpha$=0.73°C$^{-1}$ and $b$=9.63.

**APPENDIX B: ICE MUTIPLICATION FROM ICE-ICE COLLISIONS**

Three types of ice particles are considered in MIMICA: small (cloud) ice, snow, and graupel. Ice
multiplication is allowed after cloud ice-cloud ice cloud ice-snow, cloud ice-graupel, graupel-snow,
snow-snow and graupel-graupel collisions. Collisions between cloud ice-cloud ice, cloud ice-snow,
graupel-snow and snow-snow are already included in the model to represent aggregation. The rate of
number ($P_{n12}$) and mass ($P_{m12}$) concentration of particle 1 that is collected by particle 2 during these





collisions is given:


$$P_{n12} = \frac{\pi}{4}\rho E_{col}N_1 N_2 \left[ \left| D_{1n}^2 \frac{(a_1+b_{v1}+2)(a_1+b_{v1}+1)}{(a_1+2)(a_1+1)} v_{n1} - v_{n2} \right| + \left| 2D_{1n}D_{2n} \frac{(a_1+b_{v1}+1)}{(a_1+1)} v_{n1} - \frac{(a_2+b_{v2}+1)}{(a_2+1)} v_{n2} \right| + \right.$$
$$\left. D_{2n}^2 v_{n1} - \left| \frac{(a_2+b_{v2}+2)(a_2+b_{v2}+1)}{(a_2+2)(a_{12}+1)} v_{n2} \right| \right] \quad (1)$$

$$P_{q12} = \frac{\pi}{4}\rho E_{12}Q_1 N_2 \left[ \left| D_{1m}^2 \frac{(a_1+b_{v1}+b_{m1}+2)(a_1+b_{v1}+b_{m1}+1)}{(a_1+2)(a_1+1)} v_{m1} - v_{n2} \right| \right.$$
$$+ \left| 2D_{1m}D_{2n} \frac{(a_1+b_{v1}+b_{m1}+1)}{(a_1+1)} v_{m1} - \frac{(a_2+b_{v2}+b_{m1}+1)}{(a_2+1)} v_{n2} \right| + D_{2n}^2 v_{m1}$$
$$\left. - \left| \frac{(a_2+b_{v2}+2)(a_2+b_{v2}+1)}{(a_2+2)(a_{12}+1)} v_{n2} \right| \right] \quad (2)$$

where subsrcript 'n' and 'm' denote number- and mass- weighted parameters, respectively. $N$ and $Q$
refer to number and mass concentration of the particle, while $D$ and $v$ represent its diameter and
terminal velocity. $a$ is the shape parameter of the size distribution for each particle, set to 2 for cloud ice
(independently of the ice habit, 1 for snow and 0 for graupel, while $b_v$ is a coefficient in the fallspeed-
diameter relationship (see Section 3.3.1). $E_{col}$ is the collection efficiency, given as a function of
temperature (K): $E_{col=} exp[0.09(T-273.15)]$. For self-collection, thus collisions between same ice types,
the above equations take the form:

$$P_{n11} = \frac{\pi}{2}\rho E_{col}N_1 N_1 \left[ \left| D_{1n}^2 \frac{(a_1+b_{v1}+2)(a_1+b_{v1}+1)}{(a_1+2)(a_1+1)} v_{n1} \right| \right] \quad (3)$$
$$P_{q11} = \frac{\pi}{2}\rho E_{col}N_1 Q_1 \left[ \left| D_{1n}^2 \frac{(a_1+b_{v1}+2)(a_1+b_{v1}+1)}{(a_1+2)(a_1+1)} v_{n1} \right| \right] \quad (4).$$


The above equations are further used to determine collisions that result in ice multiplication, by
replacing the collection efficiency with the term $E^*=1-E_{col}$. This means that the collisions that do not
result in aggregation are those that contribute to SIP. Since aggregation after cloud-ice-graupel and
graupel-graupel collisions does not occur, we assume that 100% of these collisions result in
multiplication: $E^*=1$.

The Phillips et al. (2017a) parameterization allows for varying treatment of $F_{BR}$ depending on the ice
crystal type and habit:

$$F_{BR} = \alpha A \left( 1 - exp\left\{ -\left[ \frac{CK_o}{\alpha A} \right]^{\gamma} \right\} \right) \quad (5)$$

$K_o = \frac{m_1 m_2}{m_1+m_2}(\Delta u_{n_{12}})^2$ is the initial values of collisional kinetic energy and $a = \pi D^2$, where $D$ (in
meters) is the size of the smaller ice particle which undergoes fracturing and α is its surface area. $m_1$, $m_2$



are the masses of the colliding particles and $\Delta u_{n12}$ is the difference in their terminal velocities. A correction is further applied in $\Delta u_{n12}$ to account for underestimates when $u_{n1} \approx u_{n2}$ , following Mizuno

et al. (1990) and Reisner et al. (1998):

$$\left|\Delta u_{n_{12}}\right| = ((1.7u_{n1} - u_{n2})^2 + 0.3u_{n1}u_{n2})^{1/2}$$

$A$ represents the number density of the breakable asperities in the region of contact. $C$ is the asperity-fragility coefficient, which is a function of a correction term ($\psi$) for the effects of sublimation based on

the field observations by Vardiman (1978). Exponent $\gamma$ is a function of rimed fraction for collisions that include cloud ice and snow. Particularly, for planar ice or snow, with rimed fraction $\Psi < 0.5$, that undergoes fracturing after collisions with other ice particles:

$$A = 1.58 \cdot 10^7(1 + 100\Psi^2)\left(1 + \frac{1.33 \cdot 10^{-4}}{D^{1.5}}\right), \quad (6)$$

$$C = 7.08 \times 10^6 \psi$$

$$\psi = 3.5 \times 10^{-3}$$

$$\gamma = 0.5 - 0.25\Psi$$

For fragmentation of dendrites, $A$ and $C$ are somewhat different :

$$A = 1.41 \cdot 10^6(1 + 100\Psi^2)\left(1 + \frac{3.98 \cdot 10^{-5}}{D^{1.5}}\right), \quad (7)$$

$$C = 3.09 \times 10^6 \psi$$

$$\psi = 3.5 \times 10^{-3}$$

$$\gamma = 0.5 - 0.25\Psi$$

For graupel-graupel collisions, an explicit temperature dependency is included in the equation, while $\gamma$ is contant:

$$A = \frac{a_o}{3} + \max\left(\frac{2a_o}{3} - \frac{a_o}{9}|T - 258|, 0\right), (8)$$

$$a_o = 3.78 \cdot 10^4 \cdot \left(1 + \frac{0.0079}{D^{1.5}}\right)$$

$$C = 6.3 \times 10^6$$

$$\psi = 3.5 \times 10^{-3}$$

$$\gamma = 0.3$$

The parameterization was developed based on particles with diameters 500 μm $< D <$ 5 mm, however Phillips et al. (2017a) suggest that it can be used for particle sizes outside the recommended range as long as the input variables to the scheme are set to the nearest limit of the range. Moreover, an upper

limit for the number of fragments produced per collision is imposed, set to $F_{BR_{max}} = 100$ (Phillips et al., 2017a), for all collision types. The production rate of fragments is estimated using Eq. (1) or (3) and one of the proposed formulations for $F_{BR}$ above, e.g. $P_{BR12} = P_{n12} F_{BR}$. Whenever mass transfer also





occurs, e.g. if assume that fragments ejected from snow-gruapel collisions are added to the cloud ice category, we assume that this is only 0.1% of colliding mass (Eq. (2) or (4)) that undergoes break-up
(Phillips et al. 2017a).

**APPENDIX C: CLOUD ICE - TO - SNOW AUTOCONVERSION**

For cloud ice-to-snow autoconversion, we use the formula adapted in Wang and Chang (1993) for cloud ice-to-graupel and graupel-to-hail autoconversion:


$$P_{q\text{auto}} = Q_i \, e^{D_c\lambda} \left\{ 1 + D_c\lambda \left[ 1 + D_c\lambda \left( 0.5 + \frac{D_c\lambda}{6} \right) \right] \right\}$$

$$P_{n\text{auto}} = N_i \, e^{D_c\lambda} (1 + D_c\lambda)$$

where $\lambda = \left[ \frac{A_m \Gamma(\alpha + b_m + 1) N_i}{\Gamma(\alpha + 1) Q_i} \right]^{1/b_m}$


and $D_c$ is the critical diameter that separates the two ice cateogries. $N_i$ and $Q_i$ are the number and mass cloud ice concentrations, respectively.

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





**Tables:**


**Table 1**: Characteristic parameters in the mass-diameter ($m = a_m D^{b_m}$) and fallspeed-diameter ($v = a_v D^{b_v}$) relationships (see Section 3.3.1).

| Ice type | $a_m$ | $b_m$ | $a_v$ | $b_v$ |
|---|---|---|---|---|
| dendritic cloud ice | 0.0233 | 2.29 | 5.02 | 0.48 |
| planar cloud ice | 1.43 | 2.79 | 17.9 | 0.62 |
| snow | 0.04 | 2 | 6.962 | 0.333 |
| graupel | 65 | 3 | 199.05 | 0.8 |










**Table 2:** List of sensitivity simulations (see Section 3.3).

| Simulation | Breakup process | Ice Habit | Rimed Fraction | Ice-to-snow autoconversion critical diameter (μm) |
|---|---|---|---|---|
| CNTRLDEN | off | dendrite | – | off |
| CNTRLPLA | off | plate | – | off |
| BRDEN0.2 | on | dendrite | 0.2 | off |
| BRDEN0.3 | on | dendrite | 0.3 | off |
| BRDEN0.4 | on | dendrite | 0.4 | off |
| BRPLA0.2 | on | plate | 0.2 | off |
| BRPLA0.3 | on | plate | 0.3 | off |
| BRPLA0.4 | on | plate | 0.4 | off |
| CNTRLDENauto1 | off | dendrite | – | 125 |
| BRDEN0.2auto1 | on | dendrite | 0.2 | 125 |
| BRDEN0.4auto1 | on | dendrite | 0.4 | 125 |
| CNTRLPLAauto1 | off | plate | – | 125 |
| BRPLA0.2auto1 | on | plate | 0.2 | 125 |
| BRPLA0.4auto1 | on | plate | 0.4 | 125 |
| CNTRLDENauto2 | off | dendrite | – | 500 |
| BRDEN0.2auto2 | on | dendrite | 0.2 | 500 |
| BRDEN0.4auto2 | on | dendrite | 0.4 | 500 |
| CNTRLPLAauto2 | off | plate | – | 500 |
| BRPLA0.2auto2 | on | plate | 0.2 | 500 |
| BRPLA0.4auto2 | on | plate | 0.4 | 500 |







**Table 3:** 25[th], 50[th] (median) and 75[th] percentile of LWP and IWP timeseries. All variables are in g m[-2].


| Simulation | 25[th] perc. LWP | Median LWP | 75[th] perc. LWP | 25[th] perc. IWP | Median IWP | 75[th] perc. IWP |
|---|---|---|---|---|---|---|
| ASCOS | 52.7 | 73.8 | 89.3 | 4.2 | 7.0 | 11.4 |
| CNTRLDEN | 130.6 | 142.7 | 146.7 | 1.0 | 1.9 | 3.1 |
| CNTRLPLA | 135.0 | 143.0 | 147.7 | 0.9 | 1.8 | 2.7 |
| BRDEN0.2 | 98.2 | 104.9 | 113.2 | 3.4 | 6.3 | 7.8 |
| BRDEN0.3 | 104.2 | 110.9 | 118.1 | 2.9 | 5.0 | 8.0 |
| BRDEN0.4 | 99.3 | 107.9 | 118.8 | 3.6 | 5.3 | 7.7 |
| BRPLA0.2 | 109.9 | 109.9 | 128.9 | 2.4 | 4.8 | 6.5 |
| BRPLA0.3 | 0.4 | 96.6 | 116.1 | 0.1 | 4.2 | 8.2 |
| BRPLA0.4 | 1.6 | 66.7 | 120.8 | 0 | 0.1 | 0.5 |
| CNTRLDENauto1 | 111.4 | 130.0 | 139.3 | 2.5 | 4.0 | 5.6 |
| BRDEN0.2auto1 | 80.2 | 94.1 | 105.8 | 4.9 | 6.2 | 7.3 |
| BRDEN0.4auto1 | 78.8 | 98.5 | 105.5 | 5.8 | 6.9 | 7.4 |
| CNTRLPLAauto1 | 121.5 | 132.8 | 142.1 | 1.7 | 3.1 | 4.5 |
| BRPLA0.2auto1 | 76.5 | 89.6 | 105.9 | 4.7 | 6.1 | 7.7 |
| BRPLA0.4auto1 | 69.1 | 77.8 | 102.7 | 4.8 | 6.4 | 8.3 |
| CNTRLDENauto2 | 133.0 | 142.1 | 146.9 | 1.0 | 1.9 | 3.1 |
| BRDEN0.2auto2 | 94.6 | 104.4 | 111.6 | 3.6 | 5.3 | 7.3 |
| BRDEN0.4auto2 | 196.1 | 106.7 | 116.2 | 3.8 | 5.6 | 7.3 |
| CNTRLPLAauto2 | 132.7 | 140.6 | 147.0 | 0.9 | 1.9 | 3.0 |
| BRPLA0.2auto2 | 97.1 | 107.6 | 117.6 | 3.2 | 5.5 | 6.8 |
| BRPLA0.4auto2 | 103.9 | 110.4 | 116.3 | 3.6 | 5.0 | 7.1 |







**Figures:**

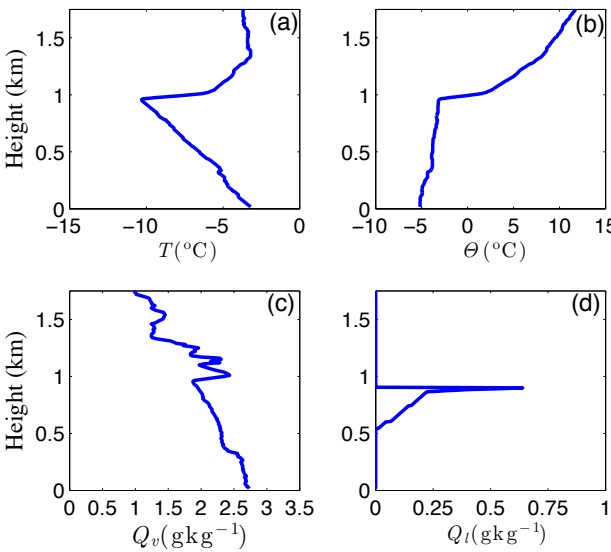

**Figure 1:** Radiosonde profiles of (a) temperature ($T$), (b) potential temperature ($\Theta$), and (c) specific humidity ($Q_v$) used to initialize the LES. The profile of cloud liquid ($Q_l$) in panel (d) is integrated from radiometer measurements.








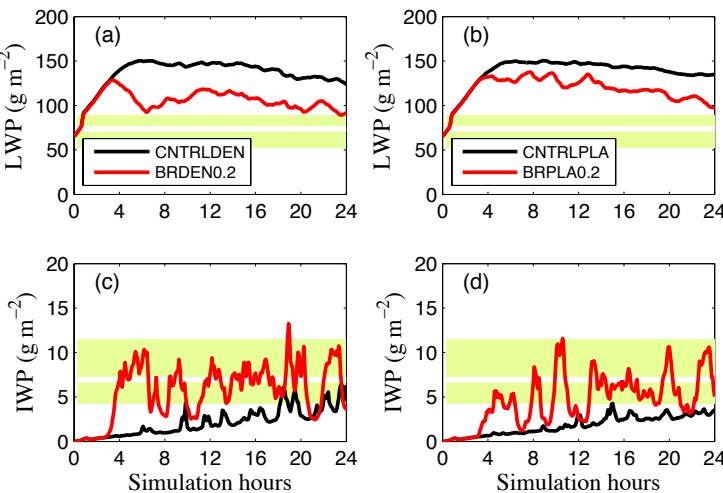

**Figure 2:** Timeseries of (a, b) LWP and (c, d) IWP for simulations with (a, c) dendrites and (b, d) plates. Light green shaded area indicates the interquartile range of observations, while the horizontal white line shows median observed values. Black lines represent simulations that account only for PIP, while red ones include the break-up process. The rimed fraction of cloud ice/snowflakes that undergo break-up is set to 0.2.









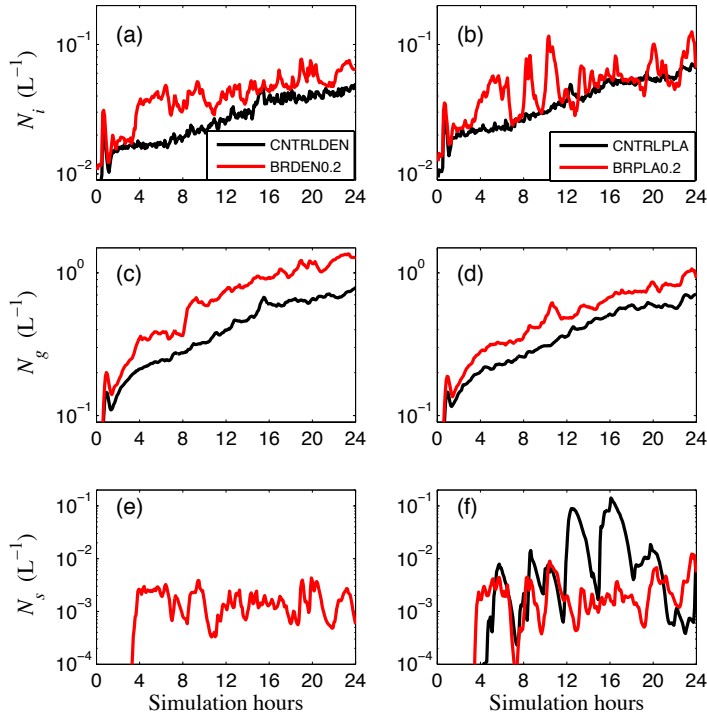

**Figure 3:** Timeseries of domain-averaged (a, b) cloud ice ($N_i$), (c, d) graupel ($N_g$) and (e, f) snow ($N_s$) number concentrations for simulations with dendrites (left column) and plates (right column). Black lines represent simulations that account only for PIP, while red ones include BR process. The rimed 005 fraction of cloud ice/snowflakes that undergo break-up is set to 0.2. Note the logarithmic y-scale.







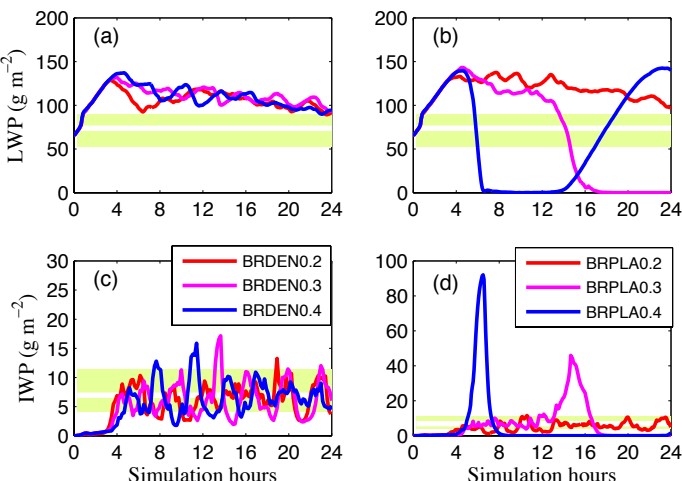

**Figure 4:** Similar to Fig. 2 but for simulations with active break-up and varying cloud ice/snow rimed fraction: 0.2 (red), 0.3 (magenta) and 0.4 (blue).







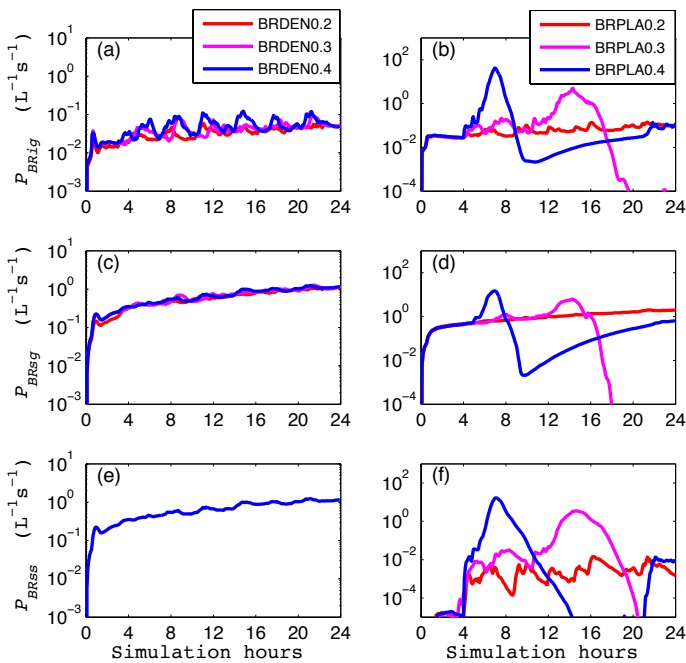

**Figure 5:** Timeseries of domain-averaged fragment generation rate ($L^{-1}s^{-1}$) from (a, b) cloud ice-graupel ($P_{BRig}$), (c, d) snow-graupel ($P_{BRsg}$) and (e, f) snow-snow collisions ($P_{BRss}$), for simulations with varying rimed fractions for cloud ice/snow: 0.2 (red), 0.3 (magenta), 0.4 (blue). Panels (a, c, e) correspond to simulations with dendrites, while (b, d, f) with planar ice. Note the logarithmic y-scale.





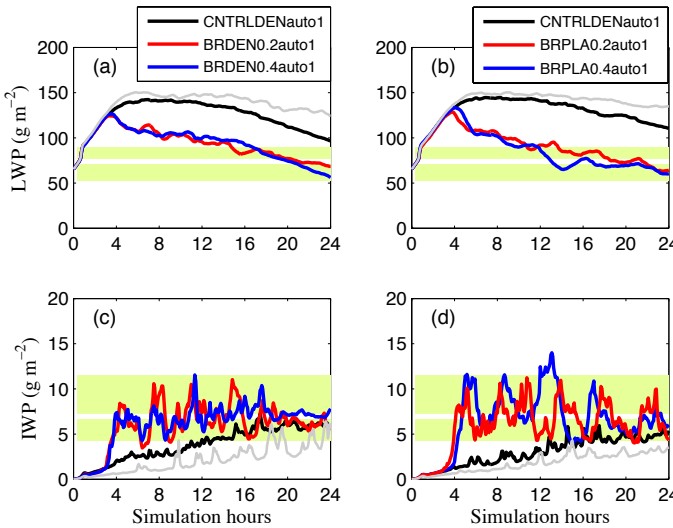


**Figure 6:** Timeseries of (a, b) LWP and (c, d) IWP for simulations with active cloud ice-to-snow autoconversion ande separation diameter set to 125 µm. The cloud ice habit is set to (a, c) dendrites and (b, d) plates. Light green shaded area indicates the interquartile range of observations, while the
horizontal white line shows median observed values. Black lines represent simulations that account only for PIP. Red lines include the break-up process with a prescribed rimed fraction for cloud ice/snow set to 0.2. Blue lines are similar to red but with the prescribed fraction set to 0.4. Light grey lines represent baseline simulations that do not account for autoconversion: (a, c) CNTRLDEN and (b, d) CNTRLPLA (see Table 2).






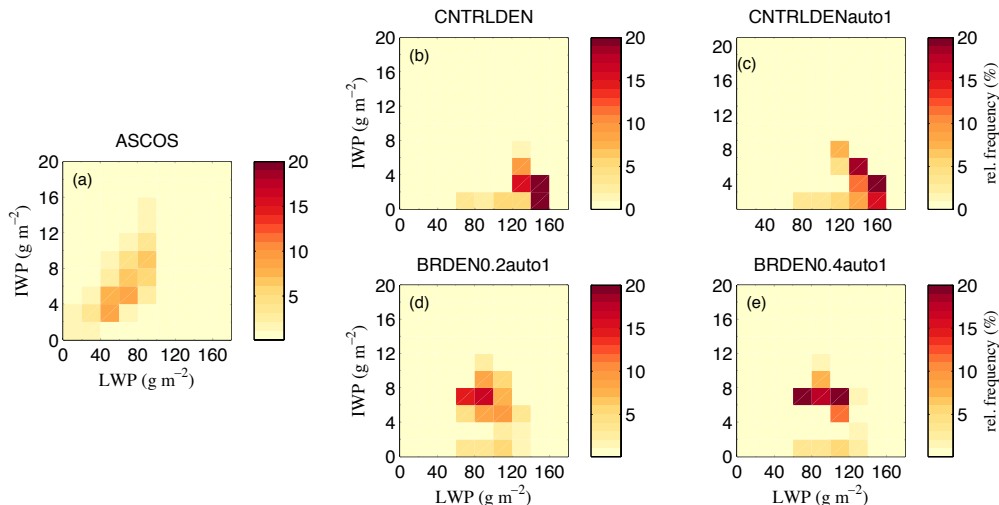

**Figure 7:** Relative frequency distribution of IWP ( g m$^{-2}$) as a function of LWP ( g m$^{-2}$) for (a) ASCOS, (b) CNTRLDEN, (c) CNTRLDENauto1, (d) BRDEN0.2auto1 and (e) BRDEN0.4auto1 (see Table 2). Cloud ice-to-snow autoconversion is active in panels (c-e). Collisional break-up is included only in panels (d-e) with the cloud ice/snow rimed fraction set to (d) 0.2 and (e) 0.4. In all simulations a dendritic cloud ice habit is assumed.







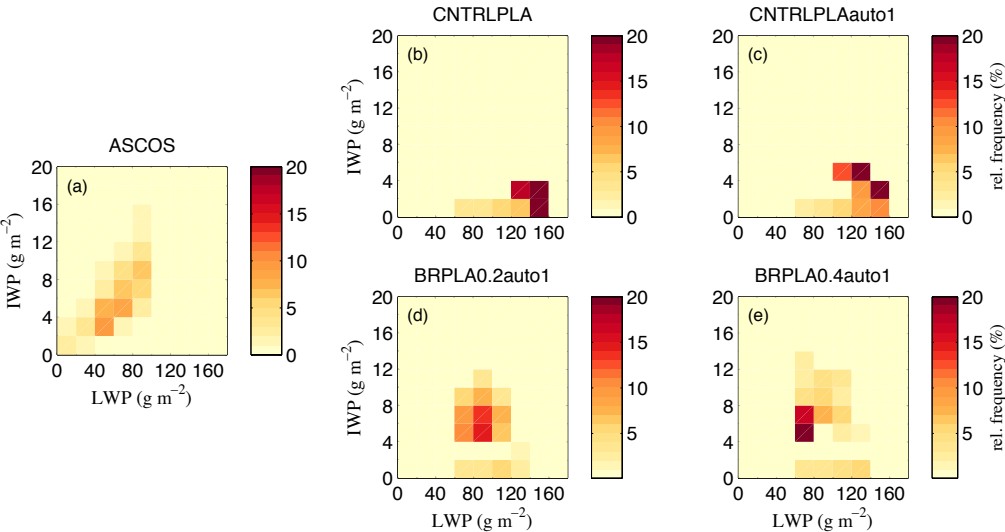

**Figure 8:** Relative frequency distribution of IWP ( g m$^{-2}$) as a function of LWP ( g m$^{-2}$) for (a) ASCOS, (b) CNTRLPLA, (c) CNTRLPLAauto1, (d) BRPLA0.2auto1 and (e) BRPLA0.4auto1 (see Table 2). The set-up in each panel is similar to Fig. 7, except that in all simulations a planar cloud ice habit is assumed.







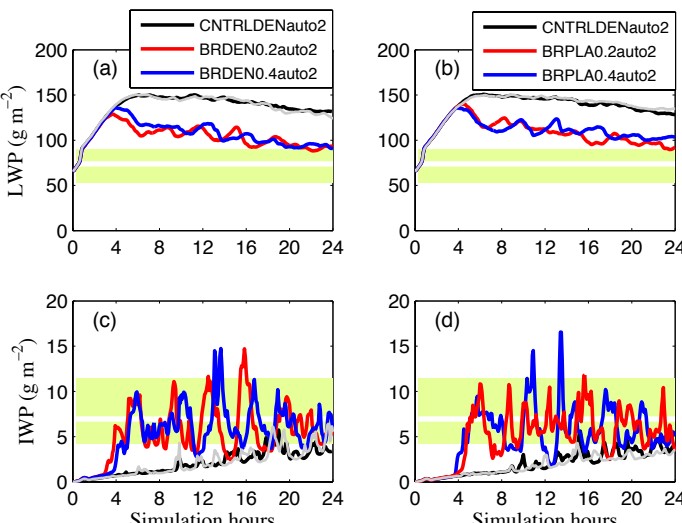

**Figure 9:** Same as Fig. 6 but for simulations that include cloud ice-to-snow autoconversion with separation diameter 500 μm (see Table 2).

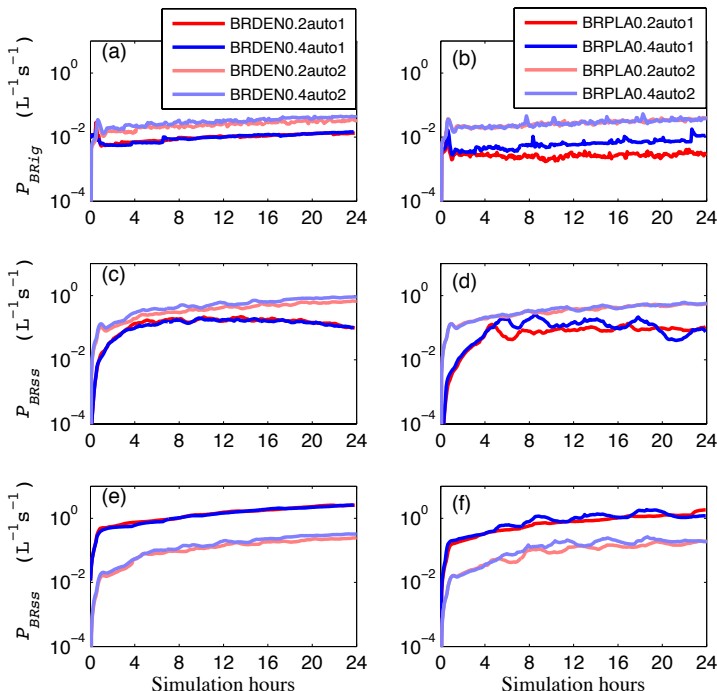

**Figure 10:** Timeseries of domain-averaged fragment generation rate ($L^{-1}s^{-1}$) from (a, b) cloud ice-graupel ($P_{BRig}$), (c, d) snow-graupel ($P_{BRsg}$) and (e, f) snow-snow collisions ($P_{BRss}$), for simulations with active cloud ice-to-snow autoconversion and collisional break-up (see Table 2). The cloud ice habit is set to (a, c) dendrites and (b, d) plates. Red (blue) lines represent simulations with a prescribed rimed fraction set to 0.2 (0.4). Darker (lighter) colors correspond to simulation with a cloud ice/snow separation diameter set to 125 μm (500 μm).