# Peer review of "Ice multiplication from ice-ice collisions in the high Arctic: sensitivity to ice habit, rimed fraction, ice type and uncertainties in the numerical description of the process"

_Atmospheric Chemistry and Physics, 2020_

## Referee Comment (RC1) · Anonymous Referee #1 · 31 Oct 2020

The manuscript titled: "Ice multiplication from ice-ice collisions in the high Arctic: sensitivity to ice habit, rimed fraction and the spectral representation of the colliding particles" by Sotiropoulou et al., discussed the impacts of adding secondary ice production (SIP) process into the LES model and its influence on Arctic cloud properties. The authors conducted several sensitivity tests to investigate the influence of ice habit, rimed fraction, and threshold value in cloud ice and snow autoconversion process on the SIP and cloud features.

General comments:

[Figure]

1. Comparison between the simulated ice number and observed ice number should be included in the study since this is the aim of this study. If the observation data for ice number is not available for this case, the author should use a different case that has this useful observation data. Otherwise, it is hard to justify if the modification in the model leading in the right direction. Lacking this comparison makes the paper less convincing to readers.

2. The scientific contribution is not significant enough for this paper. The implementation of the secondary ice production processes to the model is clearly shown in your previous paper. Just several sensitivities tests are not enough to support a whole research story. More deep analysis should be conducted, like give a physically-based explanation of changes in LWP and IWP, not only just describe the figures feature.

3. The "spectral representation" in the title and "Sensitivity to the representation of the ice particle spectrum" in Page 12 (Line 401) are confused to readers. The representation of the ice particle spectrum indicates the size distribution function, just as the authors described in the paper Line 153 "size distributions are defined by generalized Gamma functions". I think the author did a sensitivity test about the threshold value in the cloud ice and snow autoconversion process, not about the size distribution function. I suggest modifying the title and the subtitle.

Minor comments:

1. Page 3 (Line 100) what is the uncertainty range of the instrument and the observation data?

2. Page 10 (Line 325) "Planar ice is expected to generate more fragments per collision compared to plates if the diameter of the particles and the collisional kinetic energy are the same(see equations 6-7 in Appendix B). " you mean "dendrites ice is expected to generate more fragments per collision compared to plates"?

3. Page 10 (Line 309) "while the ICNC enhancement from break-up is shown in the

[Figure]

Supplementary Information (Text S2, Fig. S2)" I think a X-Y Figure (similar as Figure 2) shows the total ice enhancement is quite important, this figure show be shown in the main text. I also suggest adding a figure shows the comparison between the observed ice number and simulated ice number concentration.

4. Page 10 (Line 330) "This variability indicates that precipitation processes (i.e. the precipitation sink) are more effective". Author indicated that the decrease of cloud ice in Figure 3b is due to precipitation sink, but why the graupel number still increase in Figure 3d? considering the graupel has a larger fall speed parameter, should precipitate more quickly compared with cloud ice.

5. Page 32 (Line 1000) In Table 1, the parameters av for graupel is set to be 199.05 in the model, However, the av is usually set to be 19.3 for graupel, and is 114 for hail (Morrison et al., 2009). So, 200 seems too large for me, is any citation here to support that the Arctic graupel has big value av?

6. Page 36 (Line 1070) In Figure 2, does this mean observed LWP and IWP does not change during this time period? This figure is kind of confused, I suggested use time-series of observed LWP and IWP with uncertainty.

7. Page 36 (Line 1070) From Figure 2, the simulated LWP decreased by 50 g m–2, but IWP only increased by 5 g m–2. Does this means the total condensation is decreased? Or precipitation is increased?

8. Page 37 (Line 1095) Figure 3e does not have a black line, does this mean control simulation do not has snow?

9. Page 37 (Line 1095) Figure 3 shows graupel is the dominant ice-phase particles, it is 2 orders of cloud ice and is 3-4 orders of snow. Is that true for Arctic cloud? Graupel is the dominant ice particles in the Arctic cloud? Or it is a model dependent result? I think snow and cloud ice should have the largest fraction of total ice.

[Figure]

2020.

---

## Referee Comment (RC2) · Anonymous Referee #3 · 8 Jan 2021

**Review of "Ice multiplication from ice-ice collisions in the high Arctic: sensitivity to ice habit, rimed fraction and the spectral representation of the colliding particles" by Sotiropoulou *et al.***

**Verdict**

I recommend this paper be published after major revisions.

**Major Comments**

The results are impressive with greatly improved agreement to observations when breakup in ice-ice collisions is included. This vindicates the vision of Schwarzenboek *et al*. (2009) who made observations of this breakup occurring in Arctic clouds. It would be nice to compare the current prediction with their observations. If they measured that roughly half of all ice crystals had branches missing, is this consistent with the ice enhancement ratio of 2 measured ? Likewise with Rangno and Hobbs (2001).

There is some uncertainty in the breakup treatment. As a sensitivity test, it might be worth removing the correction factor (to correct for sublimational weakening in Vardiman's data) in the breakup scheme by Phillips et al. (2017a): what is the effect from such uncertainty ? Alternatively, if the number of fragments per collision is altered within the range of uncertainty apparent from the error-bars (a factor of 3 uncertainty) in the plots by Phillips et al., does this drastically affect the cloud simulation ?

It would be good to include a short model description perhaps near Section 3. After reading the paper, I am still unclear if MIMICA is bin or bulk microphysics and what its microphysical species are. It seems to be bulk microphysics only.

One wonders if sublimational breakup will further improve agreement with the observations when it is treated in models. If sublimation is happening in the cloud, then this might boost the breakup in ice-ice collisions by weakening the ice.

It would be good to apply the theory by Yano and Phillips (2011) to understand why the ice multiplication is weak in these Arctic clouds. You can estimate first the order of magnitude of the time for growth of snow particles to become graupel, given the typical LWC. If one replaces the "small graupel" in the theory by Yano and Phillips by "snow", then that time-scale (tau_g) gives the order of magnitude of the multiplication efficiency (c_tilde) measuring the instability of the system of ice multiplication. The average number of fragments per graupel-snow collision would be needed too.

Phillips et al. (2017b) did such estimates for their multicell convective system to estimate c_tilde and so it should be possible to do here. The authors will probably find, if they do this theoretical estimate, that the Arctic clouds are weakly unstable because the LWC is weak.

**Detailed comments**

**Abstract**

I am not sure if it is entirely accurate to say that habit and rimed fraction are "poorly constrained". Habit is something observe-able in the aircraft data (e.g. observations of axial ratio of ice particles from aircraft flights are sometimes used for model validation). Perhaps what is meant here is that most models do not have the detail required to predict these explicitly. Some models do have the detail (e.g. Hebrew University Cloud Model, which has a bin microphysics scheme with dendrites, columns etc as separate species and rimed fraction). In future work, one hopes that MIMICA can predict rimed fraction somehow.

It might be more accurate to say something to the effect that these quantities are not explicitly predicted by most cloud models currently.

Since a dendrite is a type of planar particle (axial ratio < 1), it might be more accurate to describe these two habits as "non-dendritic planar" particles and "dendrites".

1. **Introduction**

Line 56: There is a missing reference: Fu et al. is cited but not listed.

Line 59: The paper by Schwarzenboek *et al*. (2009) is by far the most important work underpinning the present study. So it needs more detail in description of how they observed breakup in the Arctic. Need to describe how they distinguished between artificial breakup on impact with the aircraft and natural breakup in the cloud before sampling.

Line 69: Where it is written "*Both studies, however, focused on relatively warm polar clouds (-3$^o$C to -8$^o$C), where rime-splintering is also active*", the impression is conveyed that the H-M process is comparable to the ice-ice collisional breakup. But when one reads the papers cited one sees it was only weakly active. Clarify.

Lines 56 and 57: Both lab/field studies by Vardiman and Takahashi et al. underpinned the Phillips et al. scheme and both involved some uncertainties. It would be a good idea to mention key issues with their experiments. For example:

- First, the particles sampled by Vardiman were on a mountainside, apparently below cloud-base, and so there was likely some sublimation before impact, which may be have weakened them. Phillips et al. (2017) had to correct for this, by adjusting the fragility coefficient inside the exponential function of the scheme. It is a large correction.
- Second, Takahashi et al. did not observe collisions between two riming particles, but rather observed a riming ice sphere colliding with an ice sphere predominantly in vapour growth (not riming). Thus, there are issues of representativeness. However, in real clouds, graupel falls in and out of zones rich in liquid, so the Takahashi-type collisions between graupel may be representative in a sense in view of the nonlinearity of ice multiplication.
- Third, we do not have observations of columns or needles breaking up, so the Phillips scheme just treats them as if they are (non-dendritic) planars. It is not ideal.

Despite such biases, Yano and Phillips (2011) argue that errors in the breakup rate per particle actually are not so important, because an explosion of ice concentration occurs anyway provided a threshold is surpassed.

Line 71:  The simulated range of in-cloud temperatures is stated.  But it is more important to know the actual cloud-top temperature of the cases.  So we are now simulating clouds with tops in the dendritic regime where we expect more fragmentation ?

**2.  Field observations**

This is fine.

**3. Ice formation in MIMICA**

This is fine.

**4. Results**

**4.1 Sensitivity to ice habit**

Line 288:  There may be a typo or error here: "*Planar ice is expected to generate more fragments per collision compared to plates if the diameter of the particles and the collisional kinetic energy are the same (see equations 6-7 …"*.   Those  two equations are for non-dendritic planars and dendrites respectively.

A plate is a special type of (non-dendritic) planar.

In this section, it needs to be mentioned that the non-dendritic planars occupy a wider range of temperatures than the dendrites (if this is so here), which boosts the impact from non-dendritic planars.

**4.2 Sensitivity to rimed fraction**

Line 358:  Why is cloud-ice supposed to have as high a rime fraction as snow ?   Riming does not start until sizes of a few hundred microns typically (PK97).   Need to denote the size range of "cloud-ice" here.

**4.3 Sensitivity to autoconversion**

What is the difference in microphysical processes that cloud-ice and snow are participating in ?   This seems to be the reason for the sensitivity of this size threshold.    I think the best treatment of this autoconversion is from Ferrier (1992) as it preserves the slope parameter when converting cloud-ice to snow.

**5. Discussion**

Line 458:  The rimed fraction noted in this sentence does not seem so low in actuality:  *"Uncertainties in ice habit are in general not important as long as a low rimed fraction (~0.2) is assumed"*.   The Phillips et

al. (2017a) scheme recommends a default value of 0.1 for the rimed fraction for snow > 1 mm being linearly interpolated to zero at sizes of 0.1 mm (cloud-ice).  They actually simulated the rime fraction in their models and 0.1 was more or less what was predicted for a cold cloud-base.

Could there be some compensation of errors among different parts of the microphysics ?   It is possible that, although MIMICA now appears to be a fine model, the current state of knowledge in laboratory observations of ice microphysics is still limited.   Any model is only as good as the empirical basis underpinning it.

Need to mention possibility of other overlooked SIP processes also playing a role in Arctic clouds.  See Field *et al.* (2017).

For example, sublimational breakup might be important for Arctic clouds, since downdrafts only need to descend by a few hundred meters to go from being water saturated to ice saturated if adiabatic with constant vapour mixing ratio.   There are other ideas, such as the notion of enhanced supersaturations in the wake of falling precipitation particles, which was mentioned at AGU this year.

Do the present results accord with aircraft observations by Schwarzenboek et al. who  published a histogram of missing branches per particle in Arctic clouds ?

**6. Conclusions**

Line 535:  Rimed fraction is noted as a poorly constrained yet very sensitive variable for the scheme.  A problem here is that it is easy to predict rimed fraction explicitly: you just include a passive scalar for the rime on snow per unit mass of air and then diagnose the rime fraction as a function of size (see Appendix Aa of Phillips et al. 2017b (Part 2)).

When will rimed fraction be predicted instead of prescribed in model development ?

**Appendix**

When the Phillips scheme is applied, is there a temporary grid of size bins constructed so as to apply the breakup scheme for each colliding bin-pair ?

---

## Author Comment (AC1) · 18 Mar 2021

Response to Reviewer 1

We are grateful to the reviewer for several constructive comments and suggestions that have helped us improve our manuscript. Reviewer's comments are given in red and our response follows in black.

1. Comparison between the simulated ice number and observed ice number should be included in the study since this is the aim of this study. If the observation data for ice number is not available for this case, the author should use a different case that has this useful observation data. Otherwise, it is hard to justify if the modification in the model leading in the right direction. Lacking this comparison makes the paper less convincing to readers.

Unfortunately measurements of cloud particle number concentrations were not measured during ASCOS. Such measurements have been collected during Arctic flight campaigns, but these are generally conducted at lower latitudes (e.g. ACCACIA, M-PACE, RACEPAC). Investigations focusing on lower-latitude clouds have been performed (e.g. Sotiropoulou et al. 2020) and indicated a possibly critical role of the examined process. However understanding microphysical interactions over the high Arctic and over multi-year ice-pack is particularly important and that is why ASCOS data (collected at ~87$^{o}$N) have extensively been used for microphysical investigations and model intercomparisons (e.g. Lowe et al 2017; Stevens et al. 2018; Christiansen et al. 2020), even though there are no detailed microphysical measurements. Thanks to previous studies, a good understanding of how different treatments of ice nucleation and CCN activation impact cloud macrophysical properties has already been established. Here we aim to build on existing knowledge and further quantify the possible impact of SIP. Furthermore, the results can be compared to previous investigations of this process, which also used macrophysical quantities to evaluate the performance of their parameterizations due to a lack of ICNC measurements (e.g. Fridlind et al. 2007; Fu et al. 2019).

2. The scientific contribution is not significant enough for this paper. The implementation of the secondary ice production processes to the model is clearly shown in your previous paper. Just several sensitivities tests are not enough to support a whole research story. More deep analysis should be conducted, like give a physically-based explanation of changes in LWP and IWP, not only just describe the figures feature.

Note that this is the first attempt to describe the process interactively in MIMICA (a parcel-model based parameterization was applied in the previous study). We believe that this work will be useful as a guide for how these processes can and should be considered in global models. Nevertheless, thank you for this comment, as it made us look into the feedbacks between ice multiplication, precipitation, changes in size distributions and sublimation in the subcloud layer more carefully. These parameters are now shown in Figures 4 and 5.

3. The "spectral representation" in the title and "Sensitivity to the representation of the ice particle spectrum" in Page 12 (Line 401) are confused to readers. The representation of the ice particle spectrum indicates the size distribution function, just as the authors described in the paper Line 153 "size distributions are defined by generalized Gamma functions". I think the author did a sensitivity test about the threshold value in the cloud ice and snow autoconversion process, not about the size distribution function. I suggest

**modifying the title and the subtitle.**

The new title is: 'Ice multiplication from ice-ice collisions in the high Arctic: sensitivity to ice habit, rimed fraction, ice type and uncertainties in the numerical description of the process'. This also refers to the new sensitivity tests that concern uncertain parameters of the break-up description, whose conduction was suggested by Reviewer 3.

**Minor comments:**
**1. Page 3 (Line 100) what is the uncertainty range of the instrument and the observation data?**

The uncertainty in LWP and IWP, i.e. the macrophysical quantities used to evaluate the results, is already stated in Section 2. We further added uncertainties in radiosonde measurements and CCN measurements, which were used to initialize the simulations. Finally, we now also state the vertical resolution for radar measurements, which indicates the uncertainty in defining cloud boundaries (cloud top and base height).

**2. Page 10 (Line 325) "Planar ice is expected to generate more fragments per collision compared to plates if the diameter of the particles and the collisional kinetic energy are the same (see equations 6-7 in Appendix B). " you mean "dendrites ice is expected to generate more fragments per collision compared to plates"?**

We apologize, this statement is wrong and has been removed. The same diameter does not imply same collisional kinetic energy, as terminal velocities are differently parameterized for the two ice habits.

**3. Page 10 (Line 309) "while the ICNC enhancement from break-up is shown in the Supplementary Information (Text S2, Fig. S2)" I think a X-Y Figure (similar as Figure 2) shows the total ice enhancement is quite important, this figure show be shown in the main text. I also suggest adding a figure shows the comparison between the observed ice number and simulated ice number concentration.**

Following the reviewer's suggestion we now have included a figure that shows the mean ICNC and IWP enhancement in the main text (and removed the corresponding figures from SI). Unfortunately there are no observations of ice number concentrations as discussed above.

**4. Page 10 (Line 330) "This variability indicates that precipitation processes (i.e. the precipitation sink) are more effective". Author indicated that the decrease of cloud ice in Figure 3b is due to precipitation sink, but why the graupel number still increase in Figure 3d? considering the graupel has a larger fall speed parameter, should precipitate more quickly compared with cloud ice.**

Thank you for spotting this, this statement was indeed wrong. Increases in cloud ice number concentration result in more cloud ice-drop collisions (thus graupel formation) and cloud ice aggregation (thus snow formation). This means that any $N_i$ decrease that follows a $N_i$ enhancement is due to cloud-ice depletion through snow and graupel formation (not through precipitation). This is why fluctuations in $N_i$ correlate with fluctuations in $N_g$ and $N_s$ in Figure S2 (which corresponds to the old Figure 3 in the previous manuscript). Due to a larger number of figures being included in the main text to study the influence of precipitation, sublimation

and particle size, we have moved this figure to the supplementary information.

**5. Page 32 (Line 1000) In Table 1, the parameters av for graupel is set to be 199.05 in the model, However, the av is usually set to be 19.3 for graupel, and is 114 for hail (Morrison et al, 2009). So, 200 seems too large for me, is any citation here to support that the Arctic graupel has big value av?**

In Milbrandt and Morrison (2013), the $a_v$ parameter is set to 62.92 for graupel particles with a density of 50 kg m$^{-3}$ (see Table 2 in their study) and 189.02 for a density of 850 kg m$^{-3}$. However in many other microphysics schemes a substantially lower $a_v$ is assumed. We could not find terminal velocity parameters specifically constrained for Arctic graupel in the literature, but since convective motions in the Arctic are weak it does make more sense to adapt the lower values.

Following the reviewer's suggestion, snow and graupel parameters in the mass-diameter and fallspeed-relationships have been replaced with those from the Morrison scheme in the revised study. While this has a negligible effect on the CNTRL simulation, it has a greater effect on ice multiplication, since fragment generation is a function of collisional kinetic energy. For moderate ice production the effect was weak. For example in BRDEN0.2 the maximum total fragment generation rate was 1.4 L$^{-1}$s$^{-1}$ while now it does not exceed 1.1 L$^{-1}$s$^{-1}$. In BRPLA0.4 however, where explosive multiplication occurs, the maximum fragment generation rate was 73.6 L$^{-1}$s$^{-1}$ in the old simulation setup while now it has decreased to 12.84 L$^{-1}$s$^{-1}$. An important impact was also found in simulations with active cloud ice-to-snow autoconversion. Enhancing snow formation results in enhanced ice multiplication; however if large terminal velocity parameters are used, the enhancement can be significantly larger. This is why adapting a low separation diameter (125 μm) for cloud ice and snow resulted in substantially more multiplication than when adapting the 500-μm threshold and thus limiting break-up of snow; note that snow-graupel collisions are the main source of fragments. In the new simulations with more moderate terminal velocities, enhancement of break-up through autoconversion results in moderate increases in fragment generation. For this reason the sensitivity of our results to the choice of the cloud ice-to-snow critical diameter has substantially decreased. This is now stated in lines 329-330, while only results for the 500-μm threshold are shown in the relevant figures.

**6. Page 36 (Line 1070) In Figure 2, does this mean observed LWP and IWP does not change during this time period? This figure is kind of confused, I suggested use time-series of observed LWP and IWP with uncertainty.**

Note that the Large Eddy Simulation does not account for changes in the large-scale forcing and aerosol conditions and thus eventually develops a cloud in a quasi-equilibrium state. In reality the 'steady' stratocumulus cloud lasted only for about twelve hours, while aerosol conditions likely changed substantially after this (Stevens et al. 2018). And even within these 12 hours vertical displacements associated with changes in the vertical large-scale forcing were observed, which cannot be captured by any LES model (see Figure 11 in Stevens et al 2011). Moreover, the model requires a relatively long spin-up time to develop its physics, so observation-model comparisons at each timestep are not very consistent. Thus LES simulations are in a sense semi-idealized. For all these reasons we use the macrophysical statistics from the

'steady' cloud layer period to evaluate our simulations (this is explained in lines 150-155). Point observations are, however, presented in the study in the RFD plots (Figures 9-10) to evaluate phase-partitioning in the model.

**7. Page 36 (Line 1070) From Figure 2, the simulated LWP decreased by 50 g m$^{-2}$, but IWP only increased by 5 g m$^{-2}$. Does this means the total condensation is decreased? Or precipitation is increased?**
Both precipitation and sublimation in the sub-cloud layer increased. The feedbacks between ice multiplication and these processes are now discussed more extensively in the revised text (Section 4.1 / Figure 4).

**8. Page 37 (Line 1095) Figure 3e does not have a black line, does this mean control simulation do not has snow?**
Yes, snow number concentrations do not exceed threshold values ($10^{-4}$ L$^{-1}$) in the CNTRL simulation. This is because snow is only treated as aggregate in the default MIMICA version and cloud ice–cloud ice collisions are not favored in the CNTRLDEN simulation. Once break-up is activated, multiplication of cloud ice results in more collisions between these particles and promotes snow formation. This is discussed in lines 372-377 in the revised text.

**9. Page 37 (Line 1095) Figure 3 shows graupel is the dominant ice-phase particles, it is 2 orders of cloud ice and is 3-4 orders of snow. Is that true for Arctic cloud? Graupel is the dominant ice particles in the Arctic cloud? Or it is a model dependent result? I think snow and cloud ice should have the largest fraction of total ice.**
Graupel can be dominant in some cases in Arctic clouds (e.g. Fitch et al. 2020), although graupel formation has been linked to the existence of convective cells in the past (Lawson and Zuidema 2009). A recent study however suggests that static destabilization through cloud-top radiative cooling can favor graupel formation in Arctic boundary-layer clouds. Nevertheless, for the examined case graupel formation is indeed a model-dependent result. Polarimetric radar measurements are not available to evaluate this behaviour. In the ASCOS intercomparison project (Stevens et al., 2018), where five models with bulk ice microphysics were compared, COSMO-LES produced only cloud ice. WRF simulated only snow, while COSMO-NWP and UM-CASIM resulted in both snow and little cloud ice, with snow being very little in the former. Note that all these models were constrained with the same primary ice production rate. MIMICA was the only model that produced graupel, however it was among the models (including UM-CASIM) that resulted in more realistic IWC values (see Figure 11 in Stevens et al. 2018), while all other models predicted very little ice content. The fact that ice type is model-dependent and the reason why MIMICA promotes riming compared to other models is now discussed extensively in lines 303-314.

**References:**

Christiansen, S., Ickes, L., Bulatovic, I., Leck, C., Murray, B. J., Bertram, A. K., et al.: Influence of Arctic microlayers and algal cultures on sea spray hygroscopicity and the

possible implications for mixed-phase clouds. *Journal of Geophysical Research: Atmospheres*, 125, e2020JD032808. https://doi.org/10.1029/2020JD032808, 2020

Fitch, Kyle E; Garrett, Timothy J.Earth and Space Science Open Archive ESSOAr; Washington, Jun 28, 2020.DOI:10.1002/essoar.10503407.1 (submitted to GRL)

Fridlind, A. M., Ackerman, A. S., McFarquhar, G., Zhang, G., Poellot, M. R., DeMott, P. J., Prenni, A. J., and Heymsfield, A. J.: Ice properties of single-layer stratocumulus during the Mixed-Phase Arctic Cloud Experiment: 2. Model results., J. Geophys. Res., 112, D24202, https://doi.org/10.1029/2007JD008646, 2007.

Fu, S., Deng, X., Shupe, M.D., and Huiwen X.: A modelling study of the continuous ice formation in an autumnal Arctic mixed-phase cloud case, Atmos. Res., 228, 77-85, https://doi.org/10.1016/j.atmosres.2019.05.021, 2019

Lawson R. P. & Zuidema P., Aircraft Microphysical and Surface-Based Radar, Observations of Summertime Arctic Clouds, *Journal of the Atmospheric Sciences, 66 (12), 3505-3529*

Loewe, K., Ekman, A. M. L., Paukert, M., Sedlar, J., Tjernström, M., and Hoose, C.: Modelling micro- and macrophysical contributors to the dissipation of an Arctic mixed-phase cloud during the Arctic Summer Cloud Ocean Study (ASCOS), Atmos. Chem. Phys., 17, 6693–6704, https://doi.org/10.5194/acp-17-6693-2017, 2017.

Stevens, R. G., Loewe, K., Dearden, C., Dimitrelos, A., Possner, A., Eirund, G. K., Raatikainen, T., Hill, A. A., Shipway, B. J., Wilkinson, J., Romakkaniemi, S., Tonttila, J., Laaksonen, A., Korhonen, H., Connolly, P., Lohmann, U., Hoose, C., Ekman, A. M. L., Carslaw, K. S., and Field, P. R.: A model intercomparison of CCN-limited tenuous clouds in the high Arctic, Atmos. Chem. Phys., 18, 11041–11071, https://doi.org/10.5194/acp-18-11041-2018, 2018.

---

## Author Comment (AC2) · 18 Mar 2021

We are grateful to the reviewer for several constructive comments and suggestions that have helped us improve our manuscript. The reviewer's comments are given in red and our response follows in black.

**Major Comments**

The results are impressive with greatly improved agreement to observations when breakup in ice-ice collisions is included. This vindicates the vision of Schwarzenboek *et al*. (2009) who made observations of this breakup occurring in Arctic clouds. It would be nice to compare the current prediction with their observations. If they measured that roughly half of all ice crystals had branches missing, is this consistent with the ice enhancement ratio of 2 measured ? Likewise with Rangno and Hobbs (2001).

We thank the reviewer for his/her comments. Rangno et al. found that about 35% of the observed ice particles have likely been produced by ice-ice collisions. This is generally consistent with the 1.5-2fold enhancement of ICNCs found in our simulations. Schwarzenboek et al. (2009) found an indication of fragmentation in 55% of their samples; however, they could confirm natural fragmentation only for 18%. The fragments generated per collision were estimated to be typically less than 5 in their study (with 1-branch crystals being more frequent). Our model predicts that only 10-12% of the particles contribute to fragmentation but a larger number of fragments (of the order of ~10) is generated per snow-graupel collision. However, Schwarzenboek et al. examined particles with sizes about 300 μm or somewhat larger. In our study, mm-size particles dominate ice multiplication. Thus, generation of more fragments per collision is expected.

A discussion on the ice particle sizes that contribute to multiplication is added in section 4.1. A qualitative comparison of the ICNC enhancement factors found in our simulations and in the results in Rangno and Hobbs (2001) is also offered in the same section, lines 421-424. Differences between our findings and Schwarzenboek et al. (2009) results are discussed in the 'Discussion' section.

There is some uncertainty in the breakup treatment. As a sensitivity test, it might be worth removing the correction factor (to correct for sublimational weakening in Vardiman's data) in the breakup scheme by Phillips et al. (2017a): what is the effect from such uncertainty ? Alternatively, if the number of fragments per collision is altered within the range of uncertainty apparent from the error-bars (a factor of 3 uncertainty) in the plots by Phillips et al., does this drastically affect the cloud simulation ?

We added sensitivity tests in which the sublimation correction factor has been removed from the parameterization. This resulted in explosive multiplication and cloud glaciation for both simulations with dendrites and plates. Activating ice-to-snow autoconversion, and thus enhancing precipitation, prevents cloud glaciation in simulations with dendrites but not for plates. These results are discussed in section 3.3.4

It would be good to include a short model description perhaps near Section 3. After reading the paper, I am still unclear if MIMICA is bin or bulk microphysics and what its microphysical species are. It seems to be bulk microphysics only.

MIMICA includes a bulk microphysics scheme, this is now explicitly stated in Section 3.1 to avoid confusion. Also, a summary of all the included ice-liquid interactions is now given in the same section, while the corresponding formulas can be found in Wang and Chang (1993).

One wonders if sublimational breakup will further improve agreement with the observations

when it is treated in models. If sublimation is happening in the cloud, then this might boost the breakup in ice-ice collisions by weakening the ice.
Examination of the domain-averaged profiles of saturation with respect to ice does not indicate subsaturated conditions within the cloud. This is now mentioned in the 'conclusions' section.

It would be good to apply the theory by Yano and Phillips (2011) to understand why the ice multiplication is weak in these Arctic clouds. You can estimate first the order of magnitude of the time for growth of snow particles to become graupel, given the typical LWC. If one replaces the "small graupel" in the theory by Yano and Phillips by "snow", then that time-scale (tau_g) gives the order of magnitude of the multiplication efficiency (c_tilde) measuring the instability of the system of ice multiplication. The average number of fragments per graupel-snow collision would be needed too. Phillips et al. (2017b) did such estimates for their multicell convective system to estimate c_tilde and so it should be possible to do here. The authors will probably find, if they do this theoretical estimate, that the Arctic clouds are weakly unstable because the LWC is weak.
We derived tau_g from two simulations, which was found to be shorter than in previous studies (7-8 min). For BRDEN0.2 and BRPLA0.2 we estimated $\hat{C}$=1.6 and $\hat{C}$=2.2 respectively. Indeed while $\hat{C}>1$, which indicates that explosive multiplication is possible, these values are substantially smaller than the value $\hat{C}$=10 estimated for a convective cloud by Phillips et al. (2017b) and for warmer Arctic clouds by Sotiropoulou et al. (2020). This is now discussed in the 'Discussion' section.

**Detailed comments**
**Abstract**
I am not sure if it is entirely accurate to say that habit and rimed fraction are "poorly constrained". Habit is something observe-able in the aircraft data (e.g. observations of axial ratio of ice particles from aircraft flights are sometimes used for model validation). Perhaps what is meant here is that most models do not have the detail required to predict these explicitly. Some models do have the detail (e.g. Hebrew University Cloud Model, which has a bin microphysics scheme with dendrites, columns etc as separate species and rimed fraction).
Since a dendrite is a type of planar particle (axial ratio < 1), it might be more accurate to describe these two habits as "non-dendritic planar" particles and "dendrites".
Thank you for this clarification. This statement has been removed from this section. We now simply discuss the fact that while most bulk microphysics schemes do not predict ice habit and rimed fraction, according to our results this is not detrimental for the representation of ice multiplication due to break-up. This is particularly important for climate models, which often employ more simplified bulk schemes (e.g. Morrison and Gettelman 2009). Finally, the term 'planar' has been replaced with 'non-dendritic planar' throughout the text.

**1. Introduction**

Line 56: There is a missing reference: Fu et al. is cited but not listed.
The reference has now been added

Line 59: The paper by Schwarzenboek *et al*. (2009) is by far the most important work underpinning the present study. So it needs more detail in description of how they observed breakup in the Arctic. Need to describe how they distinguished between artificial breakup on impact with the aircraft and natural breakup in the cloud before sampling.
We added a paragraph in the introduction that describes the results of this study:

*'Schwarzenboeck et al. (2009) found evidence of crystal fragmentation in 55% of their in-situ samples of ice particles collected with a Cloud Particle Imager during ASTAR (Arctic Study of Aerosols, Clouds and Radiation) campaign. However, natural fragmentation could only be confirmed for 18% these cases, which was identified by either subsequent growth near the break area or/and lack of a fresh break-up line (which indicates shattering on the probe). For the rest of their samples, artificial fragmentation could not be excluded. Moreover, their analysis included only crystals with stellar shape and sizes around 300 μm or roughly larger. This suggests that the frequency of collisional break-up in Arctic clouds is likely higher in reality compared to what is indicated in their study'*

Line 69: Where it is written "*Both studies, however, focused on relatively warm polar clouds (-3°C to -8°C), where rime-splintering is also active*", the impression is conveyed that the H-M process is comparable to the ice-ice collisional breakup. But when one reads the papers cited one sees it was only weakly active. Clarify.

It is now clarified that rime-splintering was weak in both studies. However, in Sotiropoulou et al. (2020) the combination of both rime-splintering and collisional break-up was essential to explain observed ICNCs, while in Sotiropoulou et al. (2021) rime-splintering had hardly any impact.

Lines 56 and 57: Both lab/field studies by Vardiman and Takahashi et al. underpinned the Phillips et al. scheme and both involved some uncertainties. It would be a good idea to mention key issues with their experiments. For example:

· _First, the particles sampled by Vardiman were on a mountainside, apparently below cloud-base, and so there was likely some sublimation before impact, which may be have weakened them. Phillips et al. (2017) had to correct for this, by adjusting the fragility coefficient inside the exponential function of the scheme. It is a large correction.

· _Second, Takahashi et al. did not observe collisions between two riming particles, but rather observed a riming ice sphere colliding with an ice sphere predominantly in vapour growth (not riming). Thus, there are issues of representativeness. However, in real clouds, graupel falls in and out of zones rich in liquid, so the Takahashi-type collisions between graupel may be representative in a sense in view of the nonlinearity of ice multiplication.

· _Third, we do not have observations of columns or needles breaking up, so the Phillips scheme just treats them as if they are (non-dendritic) planars. It is not ideal.

Thank you for all these points! These key problems regarding the Vardiman and Takahashi et al. studies are now discussed in detail in the Introduction section. The simplification regarding the treatment of column and needles as planar ice is also explicitly stated now.

Despite such biases, Yano and Phillips (2011) argue that errors in the breakup rate per particle actually are not so important, because an explosion of ice concentration occurs anyway provided a threshold is surpassed. In future work, one hopes that MIMICA can predict rimed fraction somehow. It might be more accurate to say something to the effect that these quantities are not explicitly predicted by most cloud models currently.

The explicit treatment of rimed fraction is planned as the subject of future studies. However, the general low sensitivity of our results to rimed fraction (as long as sufficient snow formation is allowed) is very encouraging regarding the representation of this process in less detailed bulk microphysics schemes. This is now discussed in the 'Discussion' and 'Conclusions' section. However we acknowledge that the explicit prediction of rimed fraction is likely critical in conditions characterized by larger multiplication efficiency of the break-up process.

Line 71: The simulated range of in-cloud temperatures is stated. But it is more important to know the actual cloud-top temperature of the cases. So we are now simulating clouds with

tops in the dendritic regime where we expect more fragmentation?

This statement is now modified to indicate the cloud-top temperature range: -9.5°C to -12.5°C. Both plates and dendrites can form in this range. -12°C is used as threshold in Phillips et al. (2017a) to separate the temperature ranges that likely favor non-dendritic or dendritic ice habits (with planar shapes being somewhat more likely).

**4. Results**
**4.1 Sensitivity to ice habit**

Line 288: There may be a typo or error here: *"Planar ice is expected to generate more fragments per collision compared to plates if the diameter of the particles and the collisional kinetic energy are the same (see equations 6-7 …"*. Those two equations are for non-dendritic planars and dendrites respectively. A plate is a special type of (non-dendritic) planar. In this section, it needs to be mentioned that the non-dendritic planars occupy a wider range of temperatures than the dendrites (if this is so here), which boosts the impact from non-dendritic planars.

Thank you, the statement was indeed wrong and has been removed. In MIMICA the characteristic parameters in mass and terminal velocity relationships remain constant throughout the simulation. This means that the ice habit remains constant and does not change as a function of temperature. However the examined temperature range is generally limited anyway as mentioned in our previous reply to a comment above.

**4.2 Sensitivity to rimed fraction**

Line 358: Why is cloud-ice supposed to have as high a rime fraction as snow ? Riming does not start until sizes of a few hundred microns typically (PK97). Need to denote the size range of "cloud-ice" here.

Indeed this is a simplification. However snow is treated as aggregate in the default MIMICA version, which means that cloud-ice can freely grow to large sizes without necessarily being converted to snow (since cloud ice-to-snow autoconversion is not treated). Offline estimates of the mean particle diameter indicated two modes in the relative frequency distribution of this parameter (Figure 1). The first one indicates small cloud particles ~200-250 μm and the second one mm-particles (found in the lower portion of the cloud). The fact that an increase in rimed fraction only affect the second mode of the distribution suggests that it is the mm-particles that contribute to collisional break-up. This mode has a comparable size to the snow category, thus the simplification of assuming the same rimed fraction for both ice types is not unreasonable. This is now discussed in section 4.1 (note we have now merged the subsections that concern ice habit and rimed fraction).

[Figure]

[Figure]

*Figure 1: Relative frequency distrubtion of the mean (a, b) cloud ice and (c, d) snow diameter for simulations with (a,c) dendrites and (b, d) plates. Purple, red and blue lines correspond to a prescribed rimed fraction of 0.1, 0.2 and 0.4 for the cloud ice and snow particles than undergo break-up.*

**4.3 Sensitivity to autoconversion**
What is the difference in microphysical processes that cloud-ice and snow are participating in? This seems to be the reason for the sensitivity of this size threshold. I think the best treatment of this autoconversion is from Ferrier (1992) as it preserves the slope parameter when converting cloud-ice to snow.
A summary of the interactions between liquid and ice particles is now offered in section 3.1. However we did find the reason behind the large sensitivity of the multiplication efficiency to the size threshold adapted for cloud-ice-to-snow autoconversion. As pointed out by reviewer 1, the characteristic parameters used for the graupel terminal velocity in the default MIMICA version are large (about one order of magnitude larger than in other stratocumulus schemes). Decreasing the $a_v$ parameter by a factor of ~10 (adapted from Morrison et al. 2005) has a negligible impact on simulations that do not account for collisional break-up. However, since collisional kinetic energy impacts the multiplication efficiency, these changes have a substantial impact on simulations with active break-up. In BRDEN0.2 the maximum total fragment generation rate was 1.4 $L^{-1}s^{-1}$ while now it does not exceed 1.1 $L^{-1}s^{-1}$. In BRPLA0.4, where explosive multiplication occurs, the sensitivity of fragment generation rate is even larger: a maximum rate of 73.6 $L^{-1}s^{-1}$ was found in the old simulation, while now it has decreased to 12.84 $L^{-1}s^{-1}$. A notable impact was also found in simulations with active cloud ice-to-snow autoconversion. Enhancing snow formation results in enhanced ice multiplication; however if large terminal velocity parameters are adapted, the enhancement can be significantly larger. This is why a low separation diameter (125 μm) for cloud ice and snow resulted in more multiplication than when adapting the 500-μm threshold and thus limiting break-up of snow; note that snow-graupel collisions are a main source of fragments (Figure 3). In the new simulations with more moderate terminal velocities, enhancement of break-up through autoconversion results in moderate increases in fragment generation. For this reason the sensitivity of our results to the choice of the cloud ice-to-snow critical diameter has substantially decreased. This is now stated in lines 321-322, while only results for the 500-μm threshold are shown in the relevant figures (note that autoconversion results are discussed in section 4.2 in the revised manuscript).

To conserve the highest moments of the ice particle spectrum, Ferrier et al. (1994) assumes that the number of cloud ice are approximately constant by converting only a few large ice crystals into snow. Thus snow formation does not prevent the accumulation of ice crystals within the cloud layer (since these are not depleted through the autoconversion process) and consequently does not prevent excessive multiplication and cloud glaciation. The simulations with the Ferrier scheme are not shown since they are similar to the runs without autoconversion; however the results are now discussed in section 4.2.

**5. Discussion**
Line 458: The rimed fraction noted in this sentence does not seem so low in actuality: *"Uncertainties in ice habit are in general not important as long as a low rimed fraction (~0.2) is assumed"*. The Phillips et al. (2017a) scheme recommends a default value of 0.1 for the rimed fraction for snow > 1 mm being linearly interpolated to zero at sizes of 0.1 mm (cloud-ice). They actually simulated the rime fraction in their models and 0.1 was more or less what was predicted for a cold cloud-base.

Note that riming is treated differently among models. This is the reason why substantial differences in the distribution of cloud ice content among the different ice types is found for different models (Stevens et al. 2018). This is now discussed in section 3.3.3. MIMICA allows graupel to form from cloud ice particles as small as 150 μm, while accretion efficiency increases with size. Nevertheless, we added simulations with a prescribed rimed fraction of 0.1; the results are very similar to the simulations with $\Psi=0.2$.

Could there be some compensation of errors among different parts of the microphysics? It is possible that, although MIMICA now appears to be a fine model, the current state of knowledge in laboratory observations of ice microphysics is still limited. Any model is only as good as the empirical basis underpinning it.
Compensation errors are common in models, so this is possible. This can be particularly true for bulk microphysics schemes, where non-physical thresholds are used to separate cloud ice, snow and graupel particles; these thresholds are often tuned differently among different schemes. However this is something that cannot be inferred from our simulation results.

Need to mention possibility of other overlooked SIP processes also playing a role in Arctic clouds. See Field *et al.* (2017). For example, sublimational breakup might be important for Arctic clouds, since downdrafts only need to descend by a few hundred meters to go from being water saturated to ice saturated if adiabatic with constant vapour mixing ratio. There are other ideas, such as the notion of enhanced supersaturations in the wake of falling precipitation particles, which was mentioned at AGU this year.
We added a paragraph regarding the potential influence of sublimation break-up and blowing snow in the discussion section:

*'Moreover, while processes like rime-splintering and drop-shattering are clearly ineffective in the examined conditions, the contribution from other SIP mechanisms has not been investigated, e.g. blowing snow and fragmentation of sublimating particles (Field et al. 2017). Sublimation of cloud ice particles can occur if cloud conditions become subsaturated with respect to ice; however a preliminary inspection of the domain-averaged supersaturation profiles did not reveal any such evidence. Furthermore, blowing snow is associated with relatively high wind speeds (Gossart et al, 2017), while during the examined ASCOS case the maximum wind speed never exceeded 5.2 m s$^{-2}$ in the boundary layer.'*

Unfortunately, currently we have no consensus about the possibility of activation of additional INPs in transient supersaturations in real cloud conditions

Do the present results accord with aircraft observations by Schwarzenboek et al. who published a histogram of missing branches per particle in Arctic clouds ?
Schwarzenboek et al. (2019) examined ice particles with sizes around 300 μm or somewhat larger and found that a maximum of ~5 fragments are generated per collision. However, they emphasize in their study that the findings are representative only for the specific flight conditions and cannot be generalized for any other ASTAR flights. Thus it is even more unlikely that these results are representative for ASCOS. In our simulations up to 13 fragments can be generated upon snow-graupel collisions, which is substantially larger than the findings in Schwarzenboek et al. (2019). However given that snow particles in MIMICA reach mm-sizes (Fig. 5), model estimations are not unreasonable. A related discussion has been added in the 'Discussion' section on lines 547-550, although no direct comparison between ASCOS simulations and ASTAR data can be conducted.

**6. Conclusions**

Line 535: Rimed fraction is noted as a poorly constrained yet very sensitive variable for the scheme. A problem here is that it is easy to predict rimed fraction explicitly: you just include a passive scalar for the rime on snow per unit mass of air and then diagnose the rime fraction as a function of size (see Appendix Aa of Phillips et al. 2017b (Part 2)). When will rimed fraction be predicted instead of prescribed in model development ?

Rimed fraction is not a very sensitive variable; simulations with dendrites give similar results independently of the prescribed rimed fraction. The only set-up that is very sensitive to rimed fraction is BRPLA0.4, thus only if highly rimed plates are assumed. This results in accumulation of many ice crystals in the cloud and eventually glaciation. But if the precipitation sink is enhanced through cloud ice-to-snow autoconversion in this set-up, the rimed fraction does not cause substantial changes in the cloud macrophysical state anymore.

The fact that our results show generally low sensitivity to the prescribed rimed fraction is positive news for larger-scale models, which employ bulk microphysics schemes that do not predict rimed fraction. Even more so, for climate model schemes like Morrison and Gettelman 2009 that do not even account for rimed particles (graupel). However, we acknowledge that this conclusion likely concerns only conditions with weak efficiency of break-up, as those examined here. Rimed fraction is expected to play a more critical role in more convective conditions and its explicit prediction is included in future model development plans. This is now discussed in the 'conclusions' section.

**Appendix**

When the Phillips scheme is applied, is there a temporary grid of size bins constructed so as to apply the breakup scheme for each colliding bin-pair?

The microphysics scheme already includes bulk descriptions for the interactions between the different ice types and within the same ice category, as aggregation is accounted for in the model. For consistency with the rest of the code, the same relationships are used to describe ice-ice collisions for ice multiplication. Thus a bulk (instead of a bin) approach is used for all processes in the model, including SIP. This is now explicitly stated at the beginning of the Appendix to avoid any confusion.

---

## Author Response (AR2)

**ANSWER TO REVIEWER**

We are grateful to the reviewer for the detailed review and the suggested improvements. The reviewer's comments are given in red, while our replies follow in black

A few minor points could be clarified. First, the paper by Phillips et al. (2017b) was not about a simulation of only one convective cloud, but rather involved a mesoscale model with a domain of 80 km x 80 km. There were many cloud-types present in the mesoscale convective system (MCS)
Thank you for this clarification, which is now inserted in the discussion section:

*Lines 554-554: " ...in previous studies of mesoscale convective systems (Phillips et al. 2017b)..."*

In section 4.3, the authors should comment about whether removal of the correction factor results in better agreement with observations.
We now clarify in the text that the removal of the correction factor results in poorer performance of the parameterization:

*Lines 482-483: "Overall, the removal of the correction factor results in poorer agreement with observations (Table 2). This indicates that while the determination of the correction factor is highly uncertain, its inclusion in the break-up parameterization is essential..."*

In the concluding section, the phrase "cloud glaciation" is vague and possibly mis-used. The term "glaciation" does not necessarily mean complete depletion of liquid. A "glaciated" cloud could have almost any amount of ice or liquid. Need to make the conclusions more quantitative.
We have replaced the term 'cloud glaciation' with 'cloud dissipation' in both abstract and 'conclusions' section to accurately describe what actually happens in our simulations.

In the concluding section, need to comment on how the "multiplication efficiency" , c, estimated in various simulations relates to the degree of explosive growth of ice concentrations. When c is only 2, does the model predict weak ice enhancement ? Vice versa?
In discussion section we know clarify that while $\hat{C}>1$ implies that explosive multiplication can happen, the required time is very long, longer than the mixing timescale of a stratocumulus cloud. For this reason a low multiplication efficiency is associated with weak ice enhancements. This is also clarified now in the 'conclusions' section:

*Lines 555-558 (Discussion): "While $\hat{C}>1$ implies that explosive multiplication is possible (Yano and Phillips 2011), the required time for this to happen is much longer than the time mixing-scale of the studied cloud. For this reason, such low $\hat{C}$ are associated with generally weak ICNC enhancement. "*

*Lines 629-630 (Conclusions): "Finally, we acknowledge that the weak influence of the rimed fraction is likely limited for conditions characterized by weak multiplication efficiency ($\hat{C}\approx2$) and thus weak ICNC enhancement, as those examined here. "*

---

## Author Response (AR3)

**RESPONSE TO THE EDITOR**

We are grateful to the editor for the very detailed examination of our manuscript. We have addressed all the corrections suggested below:

There is a very recent study: Zhao et al. ACP (2021) "Impacts of secondary ice production on Arctic mixed-phase clouds based on ARM observations and CAM6 single-column model simulations", which found that ice-ice collision is not an efficient process compared to other ice formation in the single-layer stratus with cloud-top temperatures between -10 and -15C. They also studied the relative importance of other SIP mechanisms in relation to primary ice nucleation. This work is highly relevant to your study and needs to be introduced in your paper.

Zhao et al (2021) is now referred in the introduction section  and section 3.2.2

There are a few corrections needed in the manuscript: 1. Line 61: "indentified" to "identified".

Corrected

2. Line 66: "18% these cases" to "18% of these cases"

Corrected

3. Line 538: the formula for C multiplication efficiency needs to be rewritten

thank you, subscripts are now corrected in the formula